# Kasugamycin potentiates rifampicin and limits emergence of resistance in *Mycobacterium tuberculosis* by specifically decreasing mycobacterial mistranslation

Swarnava Chaudhuri[1†], Liping Li[2†], Matthew Zimmerman[2], Yuemeng Chen[1], Yu-Xiang Chen[1], Melody N Toosky[1,3], Michelle Gardner[3], Miaomiao Pan[1], Yang-Yang Li[1], Qingwen Kawaji[1], Jun-Hao Zhu[1], Hong-Wei Su[1], Amanda J Martinot[4], Eric J Rubin[3], Veronique Anne Dartois[2*], Babak Javid[1,3*]

[1]Centre for Global Health and Infectious Diseases, Collaborative Innovation Centre for the Diagnosis and Treatment of Infectious Diseases, Tsinghua University School of Medicine, Beijing, China; [2]Public Health Research Institute, New Jersey Medical School, Rutgers, The State University of New Jersey, Newark, United States; [3]Department of Immunology and Infectious Diseases, Harvard TH Chan School of Public Health, Boston, United States; [4]Center for Virology and Vaccine Research, Beth Israel Deaconess Medical Center, Harvard Medical School, Boston, United States

**\*For correspondence:**
veronique.dartois@rutgers.edu (VAD);
bjavid@gmail.com (BJ)

[†]These authors contributed equally to this work

**Competing interests:** The authors declare that no competing interests exist.

**Abstract** Most bacteria use an indirect pathway to generate aminoacylated glutamine and/or asparagine tRNAs. Clinical isolates of *Mycobacterium tuberculosis* with increased rates of error in gene translation (mistranslation) involving the indirect tRNA-aminoacylation pathway have increased tolerance to the first-line antibiotic rifampicin. Here, we identify that the aminoglycoside kasugamycin can specifically decrease mistranslation due to the indirect tRNA pathway. Kasugamycin but not the aminoglycoside streptomycin, can limit emergence of rifampicin resistance in vitro and increases mycobacterial susceptibility to rifampicin both in vitro and in a murine model of infection. Moreover, despite parenteral administration of kasugamycin being unable to achieve the in vitro minimum inhibitory concentration, kasugamycin alone was able to significantly restrict growth of *Mycobacterium tuberculosis* in mice. These data suggest that pharmacologically reducing mistranslation may be a novel mechanism for targeting bacterial adaptation.
DOI: https://doi.org/10.7554/eLife.36782.001

## Introduction

The long treatment duration of regimens for tuberculosis are thought in part to be due to phenotypic resistance (tolerance) of a subpopulation of genetically susceptible bacteria to antibiotic-mediated killing (*Gold and Nathan, 2017*; *Maisonneuve and Gerdes, 2014*; *Lewis, 2010*; *Wakamoto et al., 2013*). A better mechanistic understanding of antibiotic tolerance is required to rationally devise regimens that may reduce tuberculosis regimen duration. Multiple mechanisms have been proposed for how mycobacteria tolerate antibiotics, including non-replicating persistence (*Saito et al., 2017*; *Gold and Nathan, 2017*), antibiotic efflux (*Adams et al., 2011*) and phenotypic variation in cell-size (*Rego et al., 2017*; *Richardson et al., 2016*). We proposed that in

**eLife digest** A bacterium called *Mycobacterium tuberculosis* is responsible for nearly 98% of cases of tuberculosis, which kills more people worldwide than any other infectious disease. This is due, in part, to the time it takes to cure individuals of the disease: patients have to take antibiotics continuously for at least six months to eradicate *M. tuberculosis* in the body.

Bacteria, like all cells, make proteins using instructions contained within their genetic code. Cell components called ribosomes are responsible for translating these instructions and assembling the new proteins. Sometimes the ribosomes produce proteins that are slightly different to what the cell's genetic code specified. These 'incorrect proteins' may not work properly so it is generally thought that cells try to prevent the mistakes from happening.

However, scientists have recently found that the ribosomes in *M. tuberculosis* often assemble incorrect proteins. The more mistakes the ribosomes let happen, the more likely the bacteria are to survive when they are exposed to rifampicin, an antibiotic which is often used to treat tuberculosis infections. This suggests that it may be possible to make antibiotics more effective against *M. tuberculosis* by using them alongside a second drug that decreases the number of ribosome mistakes.

Chaudhuri, Li et al. investigated the effect of a drug called kasugamycin on *M. tuberculosis* when the bacterium is cultured in the lab, and when it infects mice. The experiments found that Kasugamycin decreased the number of incorrect proteins assembled by the *M. tuberculosis* bacterium. When the drug was present, rifampicin also killed *M. tuberculosis* cells more efficiently. Furthermore, in the mice but not the cell cultures, kasugamycin alone was able to restrict the growth of the bacteria. This implies that *M. tuberculosis* cells may use ribosome mistakes as a strategy to survive in humans and other hosts.

When it was given with rifampicin, kasugamycin caused several unwanted side effects in the mice, including weight loss; this may mean that the drug is currently not suitable to use in humans. Further studies may be able to find safer ways to decrease ribosome mistakes in *M. tuberculosis*, which could speed up the treatment of tuberculosis.
DOI: https://doi.org/10.7554/eLife.36782.002

mycobacteria, increased specific errors in gene translation – mistranslation – led to intracellular protein variants of the drug target of rifampicin, RpoB, which resulted in phenotypic resistance to rifampicin (*Javid et al., 2014*; *Su et al., 2016*). Clinical isolates with mutations in the essential amidase *gatCAB* that mediates variation in cellular mistranslation rates had both increased mistranslation and rifampicin tolerance, suggesting that this is a clinically relevant mode of antibiotic tolerance (*Su et al., 2016*).

The indirect aminoacylation pathway is present in the majority of bacterial species (with the exception of some proteobacteria such as *Escherichia coli*), all archaea, some mitochondria and other organelles (*Sheppard and Söll, 2008*). Bacteria lacking specific glutamine and/or asparagine tRNA synthetases instead utilize a non-discriminatory glutamyl- (asparaginyl) synthetase that forms misacylated Glu-tRNA$^{Gln}$ and Asp-tRNA$^{Asn}$ aminoacyl complexes, respectively (*Curnow et al., 1997*; *Rathnayake et al., 2017*). These misacylated complexes are specifically recognized by GatCAB and amidated to the cognate Gln-tRNA$^{Gln}$ and Asn-tRNA$^{Asn}$ aminoacyl tRNAs, thereby preserving the fidelity of the genetic code. We recently identified that in mycobacteria, strains with mutations in *gatA* causing partial loss of function are not only viable, but can be isolated from patient samples (*Su et al., 2016*). These strains have much higher rates of specific mistranslation – of glutamine to glutamate, and asparagine to aspartate – since a proportion of misacylated Glu-tRNA$^{Gln}$ and Asp-tRNA$^{Asn}$ complexes are not fully converted to the cognate aminoacyl forms before taking part in translation at the ribosome. Importantly, wild-type GatCAB could also be limiting. Wild-type mycobacteria flow-sorted for lower GatCAB expression had both higher mistranslation rates and rifampicin tolerance (*Su et al., 2016*), suggesting that targeting the indirect tRNA aminoacylation pathway may present a novel and attractive means for increasing mycobacterial rifampicin susceptibility.

Here, we identify the natural product kasugamycin as a small molecule that can specifically decrease mistranslation due to the indirect tRNA aminoacylation pathway. At sub-inhibitory

concentrations, kasugamycin, but not another aminoglycoside streptomycin can increase mycobacterial rifampicin susceptibility both in vitro and in animal infection.

## Results

### Kasugamycin increases mycobacterial discrimination against misacylated tRNAs

We hypothesized that a small molecule that could specifically decrease mycobacterial mistranslation would result in increased susceptibility to rifampicin. GatCAB-mediated mistranslation is not due to ribosomal decoding errors – but rather due to misacylated Glu-tRNA$^{Gln}$ and Asp-tRNA$^{Asn}$ complexes taking part in translation (*Su et al., 2016*). In addition to other reported activities in *E. coli* (*Lange et al., 2017*; *Müller et al., 2016*; *Kaberdina et al., 2009*; *Moll and Bläsi, 2002*), the aminoglycoside kasugamycin decreased ribosomal misreading of mRNA (*van Buul et al., 1984*), but it was not known if it could also decrease errors due to translation of misacylated tRNAs, as the indirect tRNA aminoacylation pathway is not present in *E. coli*.

We tested whether kasugamycin could increase fidelity of misacylated tRNA-mediated mycobacterial mistranslation using a gain-of-function genetic reporter (*Figure 1A*). Kasugamycin at sub-inhibitory doses (*Supplementary file 1*) increased translational fidelity in both the model organism *Mycobacterium smegmatis* (Msm) and pathogenic *Mycobacterium tuberculosis* (Mtb) (*Figure 1B,C* and *Figure 1—figure supplement 1*). Importantly, kasugamycin decreased mistranslation in mycobacterial strains with mutated *gatA* that have extremely high misacylated-tRNA-mediated mistranslation due to partial loss of GatCAB function (*Su et al., 2016*) – *Figure 1D*, verifying that it was effective in increasing discrimination against translation of misacylated Glu-tRNA$^{Gln}$ and Asp-tRNA$^{Asn}$ aminoacyl-tRNAs.

To further test whether kasugamycin was able to decrease Asp-tRNA$^{Asn}$ misacylation-mediated mistranslation, we developed a hybrid cell-free translation system. A commercially available *E. coli* coupled transcription-translation system was supplemented with a non-discriminatory aspartyl-synthetase (*Ruan et al., 2008*) that was able to misacylate *E. coli* tRNA$^{Asn}$ (see Materials and methods). Addition of the non-discriminatory synthetase markedly increased mistranslation, as measured by the Nano-luciferase-GFP gain-of-function reporter. Kasugamycin, at concentrations that did not decrease GFP signal, was able to reduce mistranslation-induced Nluc signal (*Figure 1E*), confirming that kasugamycin could increase ribosomal discrimination of misacylated tRNA.

One of the activities of kasugamycin in *E. coli* is the inhibition of the translation of canonical mRNA transcripts (i.e. those with a 5' UTR including a Shine-Dalgarno sequence), but not leaderless transcripts lacking a 5' UTR (*Kaberdina et al., 2009*; *Moll and Bläsi, 2002*), although permissive translation of leaderless transcripts was not universal (*Schuwirth et al., 2006*). We wanted to test whether selective inhibition of translation of canonical but not leaderless transcripts with kasugamycin was evident in mycobacteria, especially since mycobacteria have many annotated leaderless transcripts (*Shell et al., 2015*; *Cortes et al., 2013*). We constructed a reporter strain of *M. smegmatis* that expressed two fluorescent proteins, GFP and mCherry from the same basic promoter (*Psmyc*), but the promoter driving mCherry resulted in a leaderless transcript (*Figure 1—figure supplement 2A* and Materials and methods). The translation inhibitor chloramphenicol inhibited translation of both fluorescent proteins, but kasugamycin at 1500 µg/ml, more than 10 times higher concentrations than required to decrease mistranslation, did not significantly attenuate translation of either fluorescent protein (*Figure 1—figure supplement 2B*). Kasugamycin has also been reported to generate novel 61S ribosomes in *E. coli* (*Kaberdina et al., 2009*), but our attempts to isolate such structures from *M. smegmatis* were unsuccessful (not shown).

Kasugamycin acts on the ribosome as an inhibitor of 30S initiation (*Wilson, 2014*). Could other translation inhibitors – either targeting 30S initiation or other steps of translation also decrease mistranslation of misacylated tRNAs? We tested edeine, another 30S initiation inhibitor, and chloramphenicol, an inhibitor of peptide-bond formation (*Wilson, 2014*), both at sub-inhibitory concentrations with our gain-of-function mistranslation reporter. Intriguingly, edeine and kasugamycin, but not chloramphenicol could decrease mistranslation rates (*Figure 1—figure supplement 3*), suggesting that inhibition of 30S initiation might interfere with ribosomal discrimination of misacylated tRNAs by a yet to be characterized mechanism.

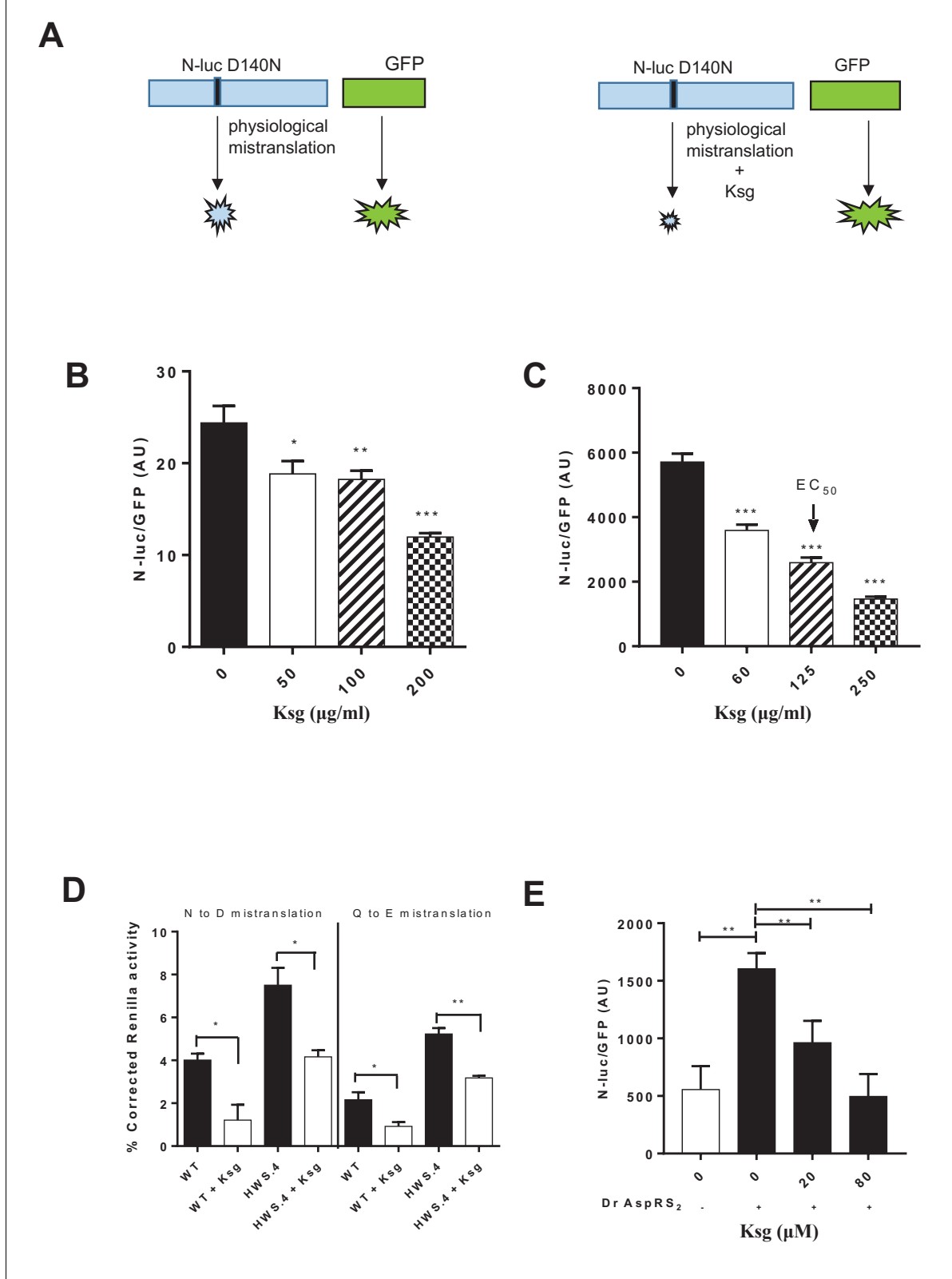

**Figure 1.** Kasugamycin decreases mycobacterial mistranslation due to the indirect tRNA aminoacylation pathway. (**A**) Schematic of the Nluc-luciferase/ GFP gain-of-function reporter to measure mistranslation. Kasugamycin (Ksg) decreases asparagine-to-aspartate mistranslation in both Msm (**B**) and Mtb (**C**) in a dose-dependent manner (see Materials and methods). (**D**) Kasugamycin (50 µg/ml) reduces asparagine-to-aspartate and glutamine-to-glutamate mistranslation – measured using a Renilla-FF luciferase dual reporter (see Materials and methods and *Su et al., 2016*) in a strain (HWS.4 – *M.*

*Figure 1 continued on next page*

*Figure 1 continued*

*smegmatis-gatA*-V405D) with a specific defect in the indirect tRNA aminoacylation pathway. (E) An *E. coli* cell-free translation system spiked with a non-discriminatory aspartyl synthetase used in conjunction with the Nluc-GFP mistranslation reporter shows that kasugamycin can specifically decrease translational error from misacylated Asp-tRNA[Asn] in a dose-dependent manner. *p<0.05, **p<0.01, ***p<0.001 by Student's t-test.

DOI: https://doi.org/10.7554/eLife.36782.003

The following figure supplements are available for figure 1:

**Figure supplement 1.** Kasugamycin decreases asparagine to aspartate and glutamine to glutamine mycobacterial mistranslation.

DOI: https://doi.org/10.7554/eLife.36782.004

**Figure supplement 2.** High-dose kasugamycin does not significantly inhibit translation of a canonical or leaderless transcript in *M. smegmatis* at doses much higher than required to reduce mistranslation.

DOI: https://doi.org/10.7554/eLife.36782.005

**Figure supplement 3.** The 30S initiation inhibitors edeine and kasugamycin decrease Asp-tRNA[Asn] mistranslation rates.

DOI: https://doi.org/10.7554/eLife.36782.006

## Kasugamycin increases mycobacterial susceptibility to rifampicin in vitro

We had previously showed that mistranslation generated via the indirect tRNA aminoacylation pathway in mycobacteria played an important role in rifampicin phenotypic resistance (*Su et al., 2016*; *Javid et al., 2014*). We therefore hypothesized that since kasugamycin could reduce mistranslation generated by this pathway, it would be able to reduce rifampicin tolerance. In keeping with our hypothesis, kasugamycin reduced mycobacterial rifampicin phenotypic resistance in wild-type (*Figure 2A,B*) and high mistranslating *gatA* mutant mycobacterial strains (*Figure 2C*). We then tested the ability of kasugamycin to enhance rifampicin-mediated killing of mycobacteria in axenic culture. Kasugamycin had no effect on mycobacterial growth, but addition to rifampicin significantly enhanced killing and sterilization of mycobacterial cultures (*Figure 2D*). Kasugamycin-mediated decrease in rifampicin tolerance was not due to its activity as an aminoglycoside – known protein translation inhibitors. The aminoglycoside streptomycin increases mistranslation (*Leng et al., 2015*). Plating of Msm on rifampicin-agar in the presence of sub-inhibitory concentrations of streptomycin increased the number of phenotypically-resistant colonies (*Figure 2—figure supplement 1*). Furthermore, in keeping with our observations with regard to 30S initiation inhibition and mistranslation, plating Msm on rifampicin-agar with sub-inhibitory concentrations of edeine but not chloramphenicol, decreased the number of phenotypically resistant colonies (*Figure 2—figure supplement 2*), confirming the link between reducing mistranslation rates and decreasing rifampicin tolerance.

To further verify that kasugamycin's effects on rifampicin susceptibility are due to its specific activity in decreasing mycobacterial mistranslation, we used a mycobacterial strain with a single point mutation in RpoB – strain Msm-RpoB-N434T. This strain has lower tolerance to rifampicin since a critical rifampicin-binding residue could no longer be mistranslated via the indirect pathway (*Su et al., 2016*). Msm-RpoB-N434T was less rifampicin tolerant than its parent strain, but kasugamycin was less potent at decreasing rifampicin tolerance further (*Figure 2E*). Since other reported activities of kasugamycin (*Lange et al., 2017*; *Müller et al., 2016*; *Kaberdina et al., 2009*) would not be affected by a single point mutation in the *rpoB* gene, we conclude that the major mechanism by which kasugamycin increased rifampicin susceptibility is by decreasing mistranslation-induced protein variants.

In *E. coli*, antibiotic resistance is preceded by tolerance (*Levin-Reisman et al., 2017*). Kasugamycin but not streptomycin given alongside rifampicin pre-treatment significantly reduced the likelihood of resistance following high-dose rifampicin challenge (*Figure 2F*), suggesting that kasugamycin may limit development of de novo resistance.

## Kasugamycin increases rifampicin susceptibility in vivo

To test activity in vivo, we needed to establish whether kasugamycin had favorable tolerability and pharmacokinetics in an animal model, and if so, was kasugamycin administration with rifampicin able to boost killing of *M. tuberculosis*. We characterized the pharmacokinetics of parenterally administered kasugamycin and streptomycin in mice. Both agents showed dose-dependent concentration-time profiles in plasma, with rapid clearance (*Figure 3—figure supplement 1*). Administration of the

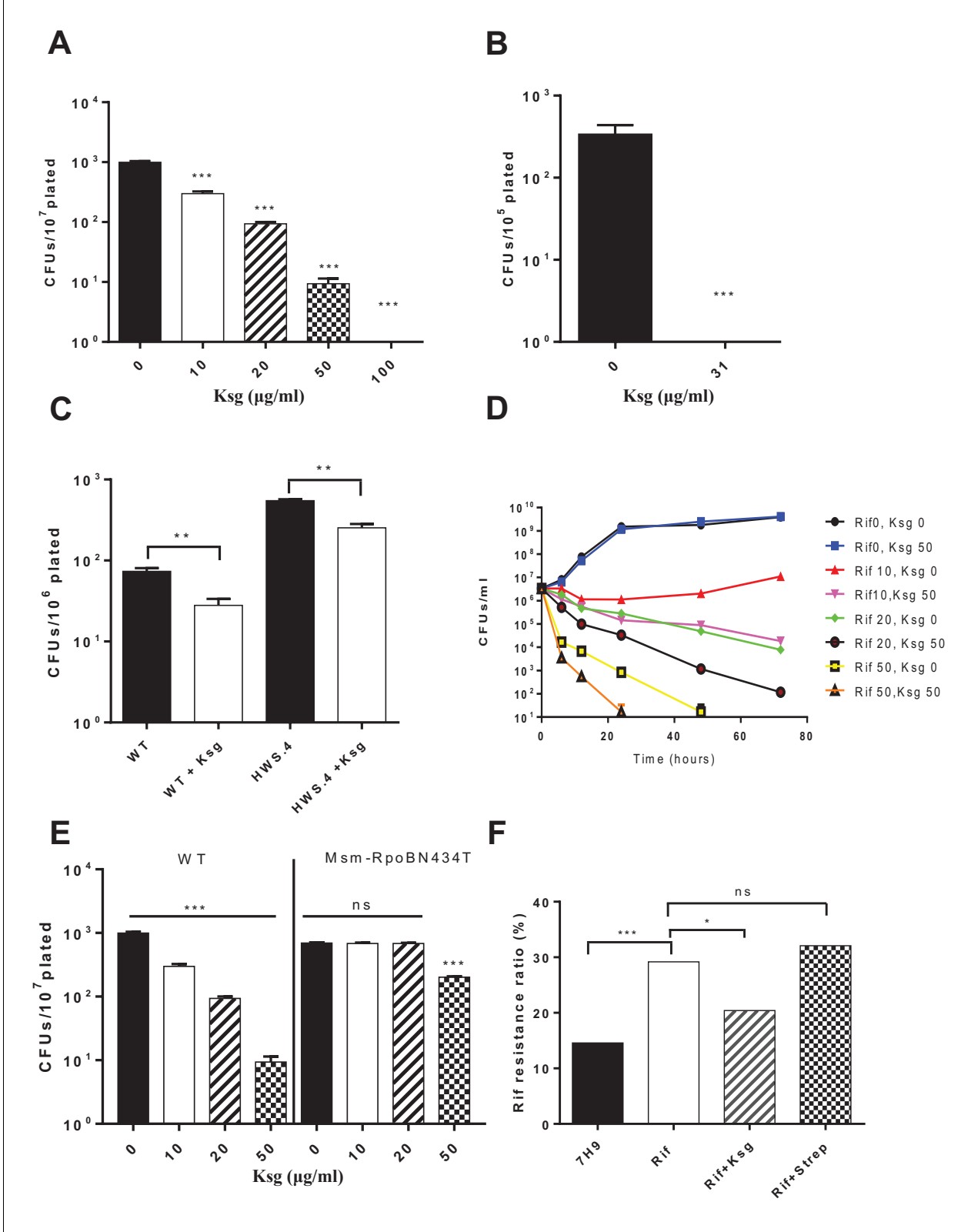

**Figure 2.** Kasugamycin enhances in vitro killing of mycobacteria by rifampicin by specifically reducing mistranslation generated by the indirect tRNA aminoacylation pathway. Ksg decreases plating rifampicin tolerance (*Su et al., 2016*) in both Msm (A) and Mtb (B) and an Msm strain with a specific defect in the indirect tRNA aminoacylation pathway (C) as well as (D) increases killing of Msm by rifampicin in axenic culture. (E) A strain (*M. smegmatis*-RpoB-N434T) with a single point mutation in *rpoB* is less tolerant to rifampicin because it is unable to mistranslate via the indirect pathway a single

*Figure 2 continued on next page*

*Figure 2 continued*

asparagine residue critical for rifampicin binding (*Su et al., 2016*). This strain is, however, relatively resistant to kasugamycin's effects on rifampicin tolerance. *p<0.05, **p<0.01, ***p<0.001 by Student's t-test. (**F**) Pre-treatment of Msm cultures with low-dose rifampicin (1 µg/ml) increases likelihood of rifampicin resistance upon subsequent challenge with high-dose rifampicin (100 µg/ml). Addition of Ksg but not streptomycin in pre-treatment abolishes the increased likelihood of rifampicin resistance. *p<0.05, ***p<0.001, ns = not significant by Fisher's exact test.

DOI: https://doi.org/10.7554/eLife.36782.007

The following figure supplements are available for figure 2:

**Figure supplement 1.** Kasugamycin but not streptomycin decreases Msm rifampicin plate tolerance.
DOI: https://doi.org/10.7554/eLife.36782.008

**Figure supplement 2.** The initiation inhibitors edeine and kasugamycin decrease Msm rifampicin plate tolerance.
DOI: https://doi.org/10.7554/eLife.36782.009

maximum tolerated daily dose of kasugamycin – 400 mg/kg – resulted in a $C_{max}$/EC(mistranslation)$_{50}$ (peak plasma concentration divided by the half-maximal effective in vitro dose of kasugamycin in reducing mistranslation rates – *Figure 1C*) of ~2.5 and time over $EC_{50}$ of 1 hr or only 4% of the dosing interval. Nevertheless, co-administration of kasugamycin resulted in an astonishing 30-fold boosting of rifampicin killing of Mtb in mouse lungs (*Figure 3A*). However, concomitant dosing of rifampicin and kasugamycin was poorly tolerated, even in the absence of tuberculosis (*Figure 3—figure supplement 2*). Histopathological examination of organs did not reveal a specific cause (not shown). Lower doses of kasugamycin co-administered with rifampicin were ineffective at enhancing rifampicin activity in vivo (not shown). In vitro, pre-treatment of axenic mycobacterial cultures with kasugamycin but not streptomycin decreased rifampicin tolerance, to a lesser degree (*Figure 3—figure supplement 3*). We thus opted for successive administration of high-dose kasugamycin intermittently with rifampicin in mice, which was well-tolerated. To specifically exclude the observed activity as being due to aminoglycoside-mediated inhibition of protein synthesis or post-antibiotic effects, we also included streptomycin-treated arms (*Figure 3B*). Since the driver of aminoglycoside efficacy is Cmax/MIC (peak plasma concentration divided by in vitro minimum inhibitory concentration) (*Scaglione and Paraboni, 2006*), we selected a streptomycin dose of 3 mg/kg, achieving a Cmax/MIC of 7, while the Cmax/MIC of kasugamycin was 0.8 (*Supplementary file 1*), thus avoiding underestimating streptomycin's activity. Sequential treatment with rifampicin and kasugamycin, but not streptomycin led to significantly enhanced killing of Mtb in mouse spleens but not lungs (*Figure 3C*, *Figure 3—figure supplement 4*), which was not explained by differences in bulk-tissue PK (*Figure 3—figure supplement 5*). Thus, despite limited drug exposure resulting in effective concentrations achieved for only a small fraction of the treatment period, the in vitro effects on rifampicin tolerance were recapitulated in mice.

## Discussion

There are multiple non-redundant models proposed for antibiotic tolerance (*Abel Zur Wiesch et al., 2015*; *Aldridge et al., 2014*; *Balaban et al., 2013*). In all these models, tolerance is mediated by generation of phenotypic heterogeneity within bacterial populations. Generation of stochastic errors during gene translation – mistranslation – is more prevalent, and occurs at higher rates, than previously appreciated (*Mohler and Ibba, 2017*; *Ribas de Pouplana et al., 2014*) and is one mechanism by which bacteria may generate considerable phenotypic heterogeneity. We show here that it is possible to identify a small molecule that can specifically decrease mistranslation rates, and hence increase antibiotic susceptibility.

There has been great interest in identifying small molecules that specifically target antibiotic-tolerant mycobacteria (*Darby and Nathan, 2010*; *Zheng et al., 2014*; *Grant et al., 2013*; *Zheng et al., 2017*; *Sukheja et al., 2017*; *Alumasa et al., 2017*; *Wang et al., 2013*; *Vilchèze et al., 2017*). These molecules have for the most part been identified via in vitro phenotypic screens that are specific for non-replicating persistence, although novel 'target-specific', whole-cell screens are proving highly useful in identifying compounds with activity against mycobacteria in vitro that are specific for certain stress-adaptation pathways (*Zheng et al., 2017*). With one exception (*Wang et al., 2013*), however, none of these candidates, and the pathways that they target, have been validated within an animal model of infection.

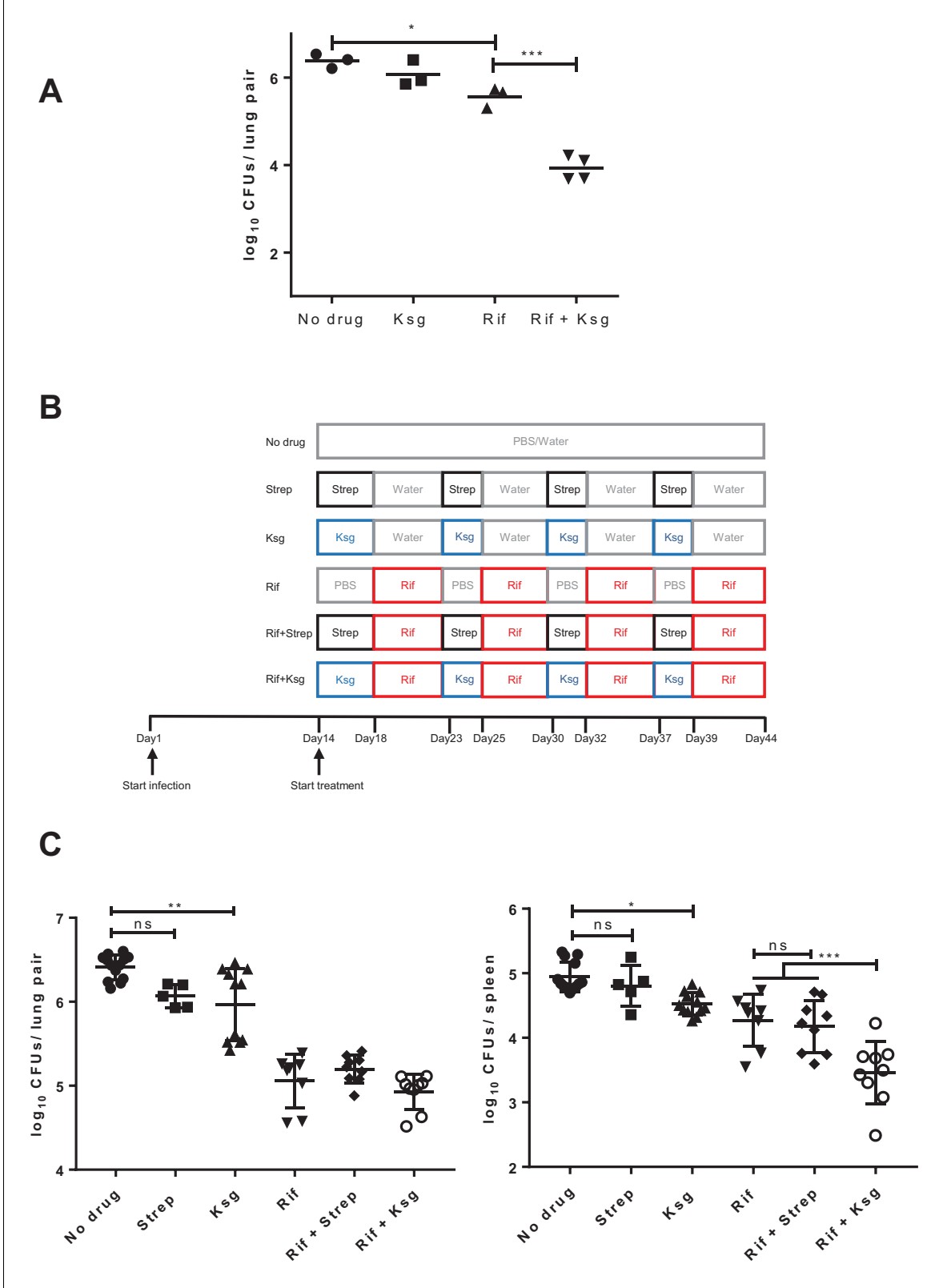

**Figure 3.** Kasugamycin enhances in vivo killing of *M.tuberculosis* by rifampicin. (**A**) Lung burden of Mtb-infected mice treated for 2 weeks with rifampicin (10 mg/kg), Ksg (400 mg/kg) or combination of the two. (**B**) Schematic of the sequential Ksg (400 mg/kg)/rifampicin (10 mg/kg) or Streptomycin (3 mg/kg)/rifampicin (10 mg/kg) dosing schedule. (**C**) Lung (left panel) or spleen (right panel) burden of Mtb-infected mice treated for 4 weeks as per the schedule in (**B**). Each data point represents bacterial organ burden from a single mouse. *p<0.05, **p<0.01, ***p<0.001, ns = not

*Figure 3 continued on next page*

*Figure 3 continued*

significant by one-way ANOVA followed by Tukey's post-hoc multi-comparison correction. Only key comparators are shown for clarity, complete statistical analysis is shown in *Supplementary file 2*.

DOI: https://doi.org/10.7554/eLife.36782.010

The following figure supplements are available for figure 3:

**Figure supplement 1.** Pharmacokinetics of intra-peritoneal administration of kasugamycin and streptomycin.

DOI: https://doi.org/10.7554/eLife.36782.011

**Figure supplement 2.** Kasugamycin-rifampicin co-administration is toxic to mice at high doses.

DOI: https://doi.org/10.7554/eLife.36782.012

**Figure supplement 3.** Kasugamycin but not streptomycin pre-treatment increases rifampicin susceptibility in Msm.

DOI: https://doi.org/10.7554/eLife.36782.013

**Figure supplement 4.** Organ burden of Mtb-infected mice treated as per the protocol in *Figure 2B*.

DOI: https://doi.org/10.7554/eLife.36782.014

**Figure supplement 5.** Tissue (lung/spleen) pharmacokinetics of kasugamycin following intra-peritoneal administration.

DOI: https://doi.org/10.7554/eLife.36782.015

Kasugamycin is unique among aminoglycosides in its ability to decrease translational error – all other aminoglycosides increase mistranslation (*Leng et al., 2015*; *Ribas de Pouplana et al., 2014*; *van Buul et al., 1984*). In addition to its known effects in reducing ribosomal decoding errors, we have shown that kasugamycin can increase discrimination against physiologically misacylated tRNAs. It had been previously demonstrated that the ribosome has some proof-reading functionality beyond Watson-Crick base-pairing. This was limited to rejection of peptides formed from incorrect codon·anti-codon base-pairs (*Zaher and Green, 2009*). Misacylated tRNAs would still form cognate codon·anti-codon pairs at the ribosome, and would therefore not be rejected by such a mechanism. Rejection of misacylated aminoacyl tRNA formation had previously been described at the aminoacyl synthetase stage (*Ibba and Söll, 1999*), or by discrimination by EF-Tu (*LaRiviere et al., 2001*). Kasugamycin's binding to the *E. coli* ribosome is close to the A794 and G926 residues (*E. coli* numbering) of 16S rRNA (*Schuwirth et al., 2006*). Given that these residues are universally conserved, it is likely that kasugamycin's binding to mycobacterial and other bacterial ribosomes is in a similar location. Therefore, our data suggest that kasugamycin-bound ribosomes also possess a hitherto unknown ability to discriminate against misacylated EF-Tu·aminoacyl-tRNA complexes. Since edeine, another 30S initiation inhibitor, but not inhibitors of elongation (streptomycin) or peptide-bond formation (chloramphenicol) could also decrease mistranslation from misacylated tRNAs, this suggests a conserved mechanism, potentially involving translation initiation, directly or indirectly, in discrimination of physiologically misacylated tRNAs.

When kasugamycin and rifampicin were given daily, there was significant potentiation in mouse lungs after 2 weeks of treatment (*Figure 3A*), but with significant toxicity. However, in the better-tolerated alternate dosing schedule (*Figure 3B*), potentiation was seen only in mouse spleens, not lungs (*Figure 3C*). These differences could not be explained by differences in kasugamycin bulk tissue distribution (*Figure 3—figure supplement 5*). Even within a single organ/tissue, there can be significant heterogeneity in drug penetration and distribution (*Prideaux et al., 2015*), which we did not measure, and such heterogeneity of distribution in either rifampicin or kasugamycin or both may explain the differences observed. Furthermore, a greater proportion of *M. tuberculosis* may be resident in macrophages in lungs compared with spleens, and aminoglycosides have poor intra-cellular penetration (*Brezden et al., 2016*). As such, under this dosing schedule kasugamycin and rifampicin may be targeting the same mycobacterial subpopulation within spleens but different subpopulations within lungs, potentially explaining the observations.

The maximum dose of kasugamycin that could be administered to mice was limited due to toxicity and pharmacokinetics. At the maximum tolerated dose (400 mg/kg, once daily), the peak plasma concentration of 300 µg/ml failed to reach the in vitro minimum inhibitory concentration of 400 µg/ml. Intriguingly, these limitations allowed us to identify the relative in vivo potency of kasugamycin compared with conventional anti-microbials. Most anti-tuberculous drugs require peak plasma concentrations orders of magnitude greater than in vitro MIC for measurable efficacy (*Mitchison, 2012*; *Pasipanodya and Gumbo, 2011*; *Pasipanodya et al., 2013*). By contrast, the bacteriostatic kasugamycin given alone, administered intermittently (10 doses in 30 days) to infected mice, and with a

Cmax/MIC <1, and at plasma concentrations <25% of MIC for 99% of the dosing interval was able to restrict Mtb growth in vivo. Streptomycin, a bactericidal aminoglycoside, given at far greater equivalent doses, had no effect. Mycobacteria increase specific mistranslation rates under conditions such as nutrient limitation and low pH (*Javid et al., 2014*), which mimic potential in vivo environments. These data suggest the efficacy of kasugamycin may not be as a conventional anti-microbial, but possibly by targeting bacterial adaptation to the host via reducing mistranslation.

There are several alternative mechanisms by which kasugamycin alone may have led to bacterial growth restriction in vivo. Kasugamycin's biological activities differ with other aminoglycosides in additional ways than its contrasting effects on translational fidelity. Most aminoglycosides inhibit translocation during protein synthesis, whereas kasugamycin inhibits translation initiation by blocking the mRNA channel in the small ribosomal subunit (*Schluenzen et al., 2006*; *Schuwirth et al., 2006*). The reported structures of kasugamycin bound to the ribosome suggest that during 70S (leaderless) initiation, there is less steric hindrance of mRNA passage than in canonical initiation (*Schluenzen et al., 2006*), potentially explaining why kasugamycin is permissive for translation of some, but not all (*Schuwirth et al., 2006*) leaderless transcripts (*Kaberdina et al., 2009*; *Moll and Bläsi, 2002*). With regard to potentiation of rifampicin in vitro, our data strongly suggest that kasugamycin is acting by reducing mistranslation generated by the indirect tRNA aminoacylation pathway, and hence protein variants that mediate rifampicin tolerance (*Figure 2E*). However, although we did not find evidence that in mycobacteria kasugamycin selectively blocks translation of canonical but not leaderless mRNA transcripts (*Figure 1—figure supplement 2*), or form alternate ribosomes (*Kaberdina et al., 2009*) (not shown), we cannot formally exclude the possibility that these mechanisms play a role in kasugamycin's activity in restricting *M. tuberculosis* growth in mice in the absence of rifampicin. Other potential mechanisms may be that kasugamycin is concentrated in macrophages, such that the intracellular concentration exceeded the MIC. Although we did not measure the intra-macrophage concentration of kasugamycin, almost all aminoglycosides enter cells poorly due to their polar structure (*Brezden et al., 2016*).

Synergistic drug combinations have the potential to radically improve treatment for tuberculosis. New methods for modelling synergy from purely empirical in vitro measurements can identify novel combinations (*Cokol et al., 2017*). More rational approaches can rescue current drugs that have limitations due to emergence of resistance or toxicity (*Blondiaux et al., 2017*). Our data suggest that targeting mycobacterial mistranslation may be a generally effective strategy, and not only in the context of potentiating rifampicin activity. Since most bacteria, with the notable exception of *E. coli* and a few other proteobacteria utilize the indirect tRNA pathway for synthesis of aminoacylated glutamine and/or asparagine tRNAs (*Curnow et al., 1997*), targeting adaptive mistranslation (*Ribas de Pouplana et al., 2014*) may be a useful strategy in the treatment of diverse bacterial infections.

## Materials and methods

### Bacterial strains, culture and antibiotics

*Mycobacterium smegmatis* mc$^2$-155 and its derivatives were cultured in Middlebrook 7H9 (BD Difco$^{TM}$) broth supplemented with 0.2% glycerol, 0.05% Tween-80, 10% Albumin-dextrose-salt (ADS), and on Luria-Bertani agar (LB agar) for plate assays. The high mistranslating strain HWS.4 with a mutation, *gatA*-V405D and the *M. smegmatis* strain with a point mutation in *rpoB* Msm-rpoB-N434T are previously described (*Su et al., 2016*). *Mycobacterium tuberculosis*-H37Rv was cultured in Middlebrook 7H9 broth with 0.2% glycerol, 0.05% Tween-80, 10% OADC (oleic acid, albumin, dextrose and catalase), and in Middlebrook 7H11 agar (BD Difco) supplemented with OADC for plate assays. Rifampicin, kasugamycin and streptomycin were purchased from Sigma. Rifampicin was dissolved in dimethyl sulphoxide (DMSO); kasugamycin and streptomycin were dissolved in water, and filter sterilized.

### Antibiotic dose selection

The in vitro MICs of kasugamycin, streptomycin and rifampicin for *M. smegmatis* and *M. tuberculosis* are given in *Supplementary file 1*. For in vitro experiments, kasugamycin and streptomycin doses were selected that had no effects on growth either in axenic culture or plating. Doses are given in the Figure legends. The only exception is *Figure 1—figure supplement 2* – testing whether

kasugamycin could inhibit translation of canonical/leaderless transcripts, when a dose of 1500 µg/ml was chose: this was 50x higher than a dose that could decrease mistranslation, and was close to the MIC. For the in vivo experiments, the maximum tolerated dose (400 mg/kg daily) of kasugamycin was given to mice. The streptomycin dose was calculated to represent 9x higher equivalent dose than kasugamycin (by Cmax/MIC) so as to not underestimate its effects, but not so high that streptomycin's known bactericidal activity might interfere with interpretation of the data. Rifampicin (10 mg/kg) is a standard dose used in most in vivo experiments.

## Generation of mistranslation reporter

The nano-luciferase gene (*nluc*) sequence was obtained from Promega and optimized to accommodate mycobacterial codon usage preference and synthesized (Genscript). The nano-luciferase gene was fused downstream of the secretion signal sequence (first forty amino acids) of the secreted mycobacterial antigen 85A/B. The fused product was PCR amplified and cloned into the pJet1.2 cloning vector (Thermo Fisher Scientific) and verified by Sanger sequencing. Site directed mutagenesis was used to create a D140N mutation in nano-luciferase, and the mutation was verified by Sanger sequencing. The mutated *nluc* gene containing the secretion signal was then cloned into the tetracycline-inducible pUVtetOR vector having a hygromycin-resistant gene cassette. After sequence verification, the recombinant plasmid was transformed into *M. smegmatis* mc$^2$-155 and *M. tuberculosis*-H37Rv using standard methodology. Codon optimized green fluorescence protein (GFP) sequence was cloned into tetracycline inducible pMC1s vector, which has kanamycin-resistant gene cassette, and electroporated into *M. smegmatis* mc$^2$-155 and *M. tuberculosis*-H37Rv containing the nano-luciferase (N-luc) reporter.

## Mistranslation assays

The basis of the N-luc/GFP assay is a gain of Nano-luciferase (N-luc) signal, similar to previously published methods (*Javid et al., 2014*; *Kramer and Farabaugh, 2007*). The D140N mutation in N-luc caused approximately 100-fold reduction in activity compared with the wild-type enzyme. Mistranslation specifically of aspartate (D) for asparagine (N) in a subset of newly translated polypeptides would result in regaining of wild-type N-luc activity. Both N-luc and GFP were induced by the same tetracycline-responsive promoter, therefore total N-luc activity was corrected by dividing by total GFP fluorescence to account for variation in gene expression between samples. The N-luc/GFP ratio gave only a relative, not absolute, indication of specific mistranslation rates.

*M. smegmatis* mc$^2$-155 containing the N-luc and GFP reporter plasmids was grown in 7H9 broth containing hygromycin (50 µg/ml) and kanamycin (20 µg/ml) to late log phase (OD$_{600}$ = 2.0) at 37°C, after which anhydrotetracycline (ATc, 50 ng/ml) was added to induce N-luc expression. The bacterial culture was then immediately aliquoted into a 96-well plate (100 µl, that is approximately 6 $\times$ 10$^7$ cells per well), different concentrations of kasugamycin or water control were added, and the cultures incubated for 16 hr at 37°C with shaking. The cultures were then transferred to a 96-well black plate and GFP fluorescence measured. The plate was then centrifuged (4000 rpm for 10 min), and supernatants transferred to a 96-well white luminescence plate. The nano-luciferase assay was performed using Nano-Glo luciferase assay kit (Promega), and luminescence measured by the same machine. Relative mistranslation rates for *M. tuberculosis* H37Rv was measured in the same way, with minor modification. The H37Rv strain containing the N-luc-D140N and GFP reporter plasmids was induced with ATc (100 ng/ml) for 2 days before measuring the N-luc and GFP signals.

Mistranslation measurement using the Renilla-Firefly dual luciferase was performed as described previously (*Su et al., 2016*). Briefly, *M. smegmatis* mc$^2$-155 strains harboring the reporters were grown till stationary phase. The cultures were diluted 20 times in fresh 7H9 medium, and expression of the dual luciferase was induced with 50 ng/ml anhydrotetracycline (ATc). After 6 hr, the bacterial cells were lysed and luciferase activities measured by dual luciferase assay kit (Promega). Measurements of fluorescence/luminescence of *M. smegmatis* were performed on a Fluoroskan Ascent FL Fluorimeter and Luminometer (Thermoscientifc), and for *M. tuberculosis* (and the Edeine/Chloramphenicol experiments) on a Biotek Synergy H1 plate reader (Fisher Scientific). The different instruments used was largely responsible for the differences in arbitrary unit (AU) values for mistranslation rates between the two species. Mistranslation rates were calculated as previously (*Su et al., 2016*). Tests of difference of means were performed by two-tailed Student's t-test.

## Cloning, expression and purification of *Deinococcus radiodurans* AspRS2 (Dr AspRS2)

The *Deinococcus radiodurans* non-discriminatory aspartyl synthetase Dr AspRS2 is able to misacylate *E. coli* asparagine tRNA with aspartate (*Ruan et al., 2008*) (i.e. form Asp-tRNA$^{Asn}$ – the mycobacterial non-discriminatory enzyme does not recognize *E. coli* tRNA, not shown), and was therefore used in the *E. coli* cell-free translation system. Codon optimized Dr AspRS2 gene containing 6xHis-tag at the 5′end was synthesized (Genewiz) and cloned into *XbaI* and *EcoRI* restriction sites of pET28a(+) vector and transformed into *E. coli* BL21(DE3). Expression of Dr AspRS2 was induced by adding 1 mM IPTG. After one hour of induction at 37°C, the cells were harvested and Dr AspRS2 was purified by Ni-NTA affinity chromatography using standard methods. The final protein concentration was determined by Bradford reagent (Bio-Rad). The activity of the enzyme was confirmed by $^3$H-Aspartate labeling of *E. coli* tRNA$^{Asn}$ (not shown).

## Cell-free measurement of mistranslation

For cell-free measurement of mistranslation, a reporter expressing mutated N-luc (D140N) linked with wild-type GFP by a GGSGGG linker was generated. The mutated *n-luc* was produced by site directed mutagenesis and linked to WT *gfp* by Gibson assembly. The *nluc* linked *gfp* was then cloned into pIVEX vector (5 Prime) for in vitro coupled transcription-translation (IVT). The reporter measured relative mistranslation of aspartate for asparagine by gain-of-function Nluc activity/GFP fluorescence, as above. The coupled transcription-translation IVT reaction was carried out using an *E. coli* T7 S30 Extract System for Circular DNA kit (Promega) following manufacturer's instructions with the following modifications. Since *E. coli* lacks the indirect tRNA aminoacylation pathway, the reaction mix was spiked with the non-discriminatory Dr AspRS2 to form misacylated Asp-tRNA$^{Asn}$ complexes that could take part in translation. 2 μM Dr AspRS2 or reaction buffer was added to the IVT reaction mix. To determine if kasugamycin could increase ribosomal discrimination of misacylated Asp-tRNA$^{Asn}$-mediated translational errors, different concentrations of kasugamycin or carrier were added to the reaction mix and incubated on ice for 10 min before addition of the reporter template. Once the DNA template was added, the tube was mixed thoroughly and incubated at 37°C for 2 hr before the reaction was quenched by placing on ice for 5 min. The Nluc activity and GFP fluorescence was measured as above.

## Inhibition of leaderless and canonical translation assay

A dual-fluorescent reporter was constructed to measure translation of leaderless and canonical transcripts. The *Psmyc* promoter was cloned from plasmid PML1357 (*Huff et al., 2010*) (Addgene) and its transcription start site mapped by 5' RACE (not shown). The gene for *mCherry* was fused directly to the transcription start site, forming a leaderless expressed gene, and *gfp* was cloned 3' to a canonical Shine-Dalgarno sequence (*Figure 1—figure supplement 2*). Both fluorescent proteins had a C-terminal tag, AAV, which decreases stability and half-life of expressed proteins by targeting them for protease-mediated degradation (*Andersen et al., 1998*), thereby allowing monitoring of translation in real time. The two expression cassettes were subcloned into plasmid PSE100 and transformed into wild-type *M. smegmatis*. The transcription start sites of the two fluorescent promoters was verified by 5'RACE and was as predicted (not shown). Biological triplicates of the strain were grown to log phase, and then back-diluted to lag phase. After 4 hr growth, antibiotics (or carrier) were added to cultures. OD$_{600}$, green and red fluorescence were measured by a Varioskan FLASH (Thermo) instrument.

## Rifampicin-specific phenotypic resistance (RSPR) assay

Rifampicin tolerance (phenotypic resistance) on agar medium was measured as previously described (*Su et al., 2016*). Stationary phase *M. smegmatis* mc$^2$-155 cultures were serially diluted, and spread on LB-agar plates containing rifampicin (50 μg/ml), with or without kasugamycin/streptomycin. The plates were incubated at 37°C for 5 – 7 days after which the number of colony-forming units (cfu) were counted. The number of bacterial cells in the inoculum was calculated by plating serial dilutions of the culture on antibiotic-free LB-agar plates and counting total plated cfu. For RSPR analysis of *M. tuberculosis* H37Rv, bacteria were spread on 7H11 agar medium containing rifampicin (0.2 μg/ml) with or without kasugamycin (31 μg/ml), and the plates were incubated for 6 weeks. For antibiotic

pre-treatment experiments, *M. smegmatis* mc$^2$-155 was grown in 7H9 broth containing kasugamycin (50 µg/ml) or streptomycin (0.25 µg/ml) and then spread on LB agar containing rifampicin. In all cases, doses of kasugamycin or streptomycin or other antibiotics were selected that did not by themselves decrease plating efficiency. Tests of difference of means were performed by two-tailed Student's t-test.

## Minimum duration of killing assay

*M. smegmatis* mc$^2$-155 was grown overnight in 7H9 (OD$_{600}$ = 0.6) broth, and approximately $5 \times 10^6$ cells were inoculated into fresh 7H9 broth containing different concentrations of rifampicin (0, 10, 20, 50 µg/ml), with or without kasugamycin (50 µg/ml). At different time points, aliquots were removed from each culture, cells were washed once and 10-fold dilutions spread onto LB agar medium with sterile glass beads. The number of viable bacteria at each time point was calculated from the resulting number of colonies.

## Rifampicin resistance assay

*M. smegmatis* mc$^2$-155 was grown to early stationary phase in 7H9 medium containing DMSO, rifampicin (1 µg/ml), or rifampicin with kasugamycin (50 µg/ml) or streptomycin (0.2 µg/ml) for 3 hr. After pre-exposure, cells were washed twice in PBS and 100 µl aliquots ($2.5 \times 10^7$ cells) were transferred to wells in 96-well plates (240 wells/group in total, experiments conducted with three independent cultures over three separate days and results pooled) containing 7H9 and rifampicin (100 µg/ml), which was selective for *bona fide* genetic resistance to the antibiotic. After 4 days of incubation, the number of rifampicin resistant cultures was observed by the presence of turbid growth. The 'resistance ratio' was calculated as the percentage of wells with turbid (resistant) cultures divided by the total number of wells per condition. Statistical analysis was performed by Fischer's exact test (Graph-Pad Prism) to compare conditions.

## Pharmacokinetic (PK) analysis of kasugamycin and streptomycin

All mouse experiments were approved by the Institutional Animal Care and Use Committee of the New Jersey Medical School, Rutgers University, Newark, NJ, under protocol number 15114D1018. For PK analysis, 8- to 10-week-old female BALB/c mice were injected intramuscularly with 100 mg/kg, 200 mg/kg or 400 mg/kg kasugamycin dissolved in 0.9% saline. Separate analysis for single-dose injections confirmed that intra-peritoneal and intra-muscular injection had exactly the same PK profiles (not shown). Blood was collected by tail snip after 15 and 30 mins, 1, 3, 5 and 8 hr following kasugamycin injection. Plasma was separated by centrifuging the blood at 5000 rpm for 5 min. Of plasma sample, 10 µl was extracted by adding 10 µl of acetonitrile/water (1:1) and 100 µl of acetonitrile: methanol (1:1) containing 10 ng/ml of verapamil (used as internal standard to correct for differences in injection volume that may happen during HPLC run), and plasma kasugamycin levels determined by high-pressure liquid chromatography coupled with tandem mass spectrometry (LC/MS/MS). The LC/MS/MS was performed on a Sciex Applied Biosystems Qtrap 4000 triple quadrupole mass spectrometer coupled to an Agilent 1260 HPLC system. Chromatography for kasugamycin was performed on a Cogent Diamond Hydride column (2.1 × 50 mm, particle size 4 µm) using a normal phase gradient elution. The gradient used 0.1% formic acid in Milli-Q deionized water and 0.1% formic acid in acetonitrile. Kasugamycin and verapamil were ionized using ESI-positive mode ionization and monitored using masses 380.17/112.10 and 455.4/156.2, respectively. Standard curve and quality control solutions were created by diluting 1 mg/ml of DMSO stocks of kasugamycin in acetonitrile/water (1:1). 10 µl of each dilution was added to 10 µl of drug-free plasma (Bioreclamation) and 100 µl of acetonitrile: methanol (1:1) containing 10 ng/ml verapamil. These standards were extracted as mentioned above.

For PK analysis of streptomycin, (8-10) week-old female BALB/c mice were intraperitoneally injected with 10 mg/kg, 20 mg/kg and 50 mg/kg of streptomycin dissolved in 0.9% saline, and blood was collected by tail snip after 15 and 30 mins, 1, 3, 5 and 8 hr. To 10 µl of plasma samples, 15 µl of extraction solution (35% trichloroacetic acid in water) was added. Then 10 µl of internal standard (20 µg/ml amikacin in water) and 70 µl of water was added to the mixture, and centrifuged at 3000 rpm for 5 min at 10℃. Of the extract, 70 µl was transferred to an analysis plate, and 3 µl was injected on

LC/MS/MS for analysis. Agilent Zorbax SB-C8, 4.6 × 75 mm, 3.5 µm was used as liquid chromatography column and Sciex Applied Biosystems Qtrap 4000 mass spectrometer was used for analysis.

### Antibiotic toxicity in mouse

This experiment was approved by the Institutional Animal Care and Use Committee of Tsinghua University under protocol number 17-BJ2. Six-week-old female BALB/c mice were given 10 mg/kg rifampicin orally by gavage and/or different doses (50 mg/kg and 400 mg/kg) of kasugamycin intra-peritoneally daily for a week. Bodyweights of the mice were monitored daily. Mice were euthanized if they showed signs of visible distress or if they lost >20% of initial bodyweight.

### Mouse infection and treatment

All mouse infection and treatment experiments were approved by the Institutional Animal Care and Use committee of Rutgers University. Initial sample size calculation was calculated to have 80% power to detect a 0.5 log CFU reduction based on inter-animal variability established through a large number of similar animal efficacy experiments. For the subsequent animal experiments, sample size was determined from preliminary data. Nine-week-old female BALB/c mice were infected with *M. tuberculosis* H37Rv with a Glas-Col inhalation system. An inoculum of $1 \times 10^7$ cfu/ml bacteria was added to the nebulizer, and the initial bacterial lung load was determined by sacrificing four mice after 3 hr of infection. After 1 week, four mice were sacrificed to determine the bacterial burden in the lungs before start of treatment. The mice were treated daily with 10 mg/kg rifampicin orally (po) alone, or with 400 mg/kg kasugamycin intra-peritoneally (ip) alone, or with a combination of rifampicin (10 mg/kg) and kasugamycin (400 mg/kg) or carrier controls for 2 weeks. The mice were then sacrificed, the lungs harvested and homogenized; and the homogenates spread on Middlebrook 7H11 agar supplemented with OADC.

For the intermittent rifampicin-kasugamycin treatment procedure (*Figure 2B*), 10- to 12-week-old female BALB/c mice were infected with *M. tuberculosis* H37Rv, with $1 \times 10^7$ cfu/ml added to the nebulizer, and the lung bacterial load was determined by sacrificing four mice after 3 hr of infection. After 2 weeks of infection, four mice were sacrificed to determine bacterial burden in the lungs before start of treatment. The mice were given the following antibiotic regimen: The control group received PBS ip for 4 days, then water po for 5 days. From there on, they received PBS for 2 days followed by water for 5 days; and this cycle was repeated until the end of the treatment. This treatment procedure was used for the kasugamycin and streptomycin groups as well, with PBS being replaced by 400 mg/kg kasugamycin and 3 mg/kg streptomycin, respectively. The same procedure was also applied for the rifampicin only group, with water being replaced by 10 mg/kg rifampicin. Mice that received two antibiotics were given 400 mg/kg kasugamycin or 3 mg/kg streptomycin for 4 days, and then 10 mg/kg rifampicin for 5 days. This was followed by kasugamycin or streptomycin for 2 days and rifampicin for 5 days; and the cycle repeated until the end of the experiment. After 44 days of infection, mice were euthanized, lungs and spleens of the mice were harvested and homogenized, and the homogenates were spread on Middlebrook 7H11 agar supplemented with OADC. Statistical analysis of differences in means between groups was performed by one-way ANOVA followed by Tukey's post-hoc correction for multiple samples (GraphPad Prism).

### Statistical analysis

Tests of significance are stated in the relevant methods sections and figure legends. All experiments were performed at least three times independently except for the edeine/chloramphenicol experiments (performed twice independently) and animal experiments. *Figure 2A* is representative of an experiment performed twice. For *Figure 2B*, data from two independent experiments are presented except for the Streptomycin arm, which was performed once.

## Acknowledgements

This work was in part funded by grants from the Bill and Melinda Gates Foundation (OPP1109789) and from the National Natural Science Foundation of China (31570129) to BJ. BJ is an Investigator of the Wellcome Trust (207487/B/17/Z). The Edeine was generously provided as a kind gift by Prof. Ian Brierley, University of Cambridge. We thank Martin Gengenbacher for helpful discussion regarding analysis of the data and Jiazi Wang, Jansy Sarathy and Jiye Yin for technical assistance.

## Additional information

### Funding

| Funder | Grant reference number | Author |
|---|---|---|
| Bill and Melinda Gates Foundation | OPP1109789 | Babak Javid |
| Wellcome | 207487/B/17/Z | Babak Javid |
| National Natural Science Foundation of China | 31570129 | Babak Javid |

The funders had no role in study design, data collection and interpretation, or the decision to submit the work for publication.

### Author contributions

Swarnava Chaudhuri, Formal analysis, Investigation, Methodology, Writing—original draft; Liping Li, Matthew Zimmerman, Melody N Toosky, Miaomiao Pan, Investigation, Methodology; Yuemeng Chen, Michelle Gardner, Investigation; Yu-Xiang Chen, Jun-Hao Zhu, Resources, Investigation; Yang-Yang Li, Hong-Wei Su, Resources; Qingwen Kawaji, Methodology; Amanda J Martinot, Formal analysis; Eric J Rubin, Formal analysis, Supervision, Writing—review and editing; Veronique Anne Dartois, Formal analysis, Supervision, Methodology, Project administration, Writing—review and editing; Babak Javid, Conceptualization, Data curation, Formal analysis, Supervision, Funding acquisition, Investigation, Methodology, Writing—original draft, Project administration, Writing—review and editing, Conceived the study

### Author ORCIDs

Eric J Rubin (iD) http://orcid.org/0000-0001-5120-962X
Veronique Anne Dartois (iD) https://orcid.org/0000-0001-9470-5009
Babak Javid (iD) http://orcid.org/0000-0002-6354-6305

### Ethics

Animal experimentation: All mouse infection and treatment experiments were approved by the Institutional Animal Care and Use committee of Rutgers University and mouse toxicity studies were approved by the Institutional Animal Care and Use Committee of Tsinghua University under protocol number 17-BJ2.

### Decision letter and Author response

Decision letter https://doi.org/10.7554/eLife.36782.023
Author response https://doi.org/10.7554/eLife.36782.024

## Additional files

### Supplementary files

• Supplementary file 1. Minimum inhibitory concentration (MIC) and Cmax/MIC (maximal plasma concentration divided by MIC) of compounds used in the in vivo study.
DOI: https://doi.org/10.7554/eLife.36782.016

• Supplementary file 2. Detailed statistical analysis of data presented in *Figure 2C*
DOI: https://doi.org/10.7554/eLife.36782.017

• Supplementary file 3. Oligonucleotides used in this work
DOI: https://doi.org/10.7554/eLife.36782.018

• Transparent reporting form
DOI: https://doi.org/10.7554/eLife.36782.019

## Data availability

All data generated or analysed during this study are included in the manuscript and supporting files.

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
