## [Decision Letter]

[Editors’ note: a previous version of this study was rejected after peer review, but the authors submitted for reconsideration. The first decision letter after peer review is shown below.]

Thank you for submitting your work entitled "Kasugamycin potentiates rifampicin in *Mycobacterium tuberculosis* by specifically decreasing mycobacterial mistranslation" for consideration by *eLife*. Your article has been reviewed by 4 reviewers including Madhukar Pai as the Reviewing Editor and Reviewer #1, and the evaluation has been overseen by Gisela Storz as the Senior Editor. The following individuals involved in review of your submission have agreed to reveal their identity: Andreas Diacon (Reviewer #3).

As you can see below, we received divergent feedback from the peer reviewers (reviewer #4, in particular, has raised several major concerns), and our final decision was reached after consultation between the reviewers. Based on these discussions and the individual reviews below, we regret to inform you that your work will not be considered further for publication in *eLife*.

*Reviewer #1:*

I am not a basic TB scientist, and my comments are focused more on the relevance of this study and its potential implications for TB treatment.

Current TB treatment is long, and therapy for drug-resistant TB is particularly long, toxic, and expensive. Success rates are only about 50% and identification of new TB drug targets is a key priority for the TB field.

In this novel study, the authors have identified kasugamycin as a small molecule that can specifically decrease mistranslation due to the indirect tRNA aminoacylation pathway. This, in turn could limit emergence of rifampicin resistance in vitro and increased mycobacterial susceptibility to rifampicin both in vitro and in a murine model of infection.

The study opens a potential approach to saving rifampicin, which is one of the best TB drugs we have today. Of course, much more follow-up and clinical work in humans is necessary to follow up on this proof of concept work.

As regards the lab methods, I will defer to expert reviewers working in this area.

*Reviewer #2:*

Overall this is a very nice article trying to decipher the action of kasugamycin and its synergist effects with rifampicin. Considering the high death toll of Tuberculosis and the danger of MDR and XMDR Tb strains this is an important field of wide interest.

In order to provide more context for the general reader I would suggest to expand the discussion on other approaches to combat MDR TBR from a mere list of papers to a concise explanation of alternative strategies, as little as one sentence may be sufficient. I couldn't help noticing that recent excellent work from the Baulard group (ie. Blondiaux et al., 2017) could be mentioned.

The conclusions by the authors appear to be justified, however, I feel that the statistical treatment of a sometimes limited set of experiments needs to be explained in more detail. At several cases it is not clear how many independent measurements were taken to calculate the reported mean values and the error bars. I would suggest that the authors state that explicitly including the experiments in the supplementary material. I can only assume that the figures with individual data points refer to individual mice, so the authors should clarify that.

Given that the paper is not very long, I would also suggest that the authors consider which of the figures on the supplementary material should be in the main text.

In order to provide more molecular insight I'd like to suggest that the authors discuss the actual binding site of kasugamycin in the ribosome. This may provide further evidence for the hypothesis put forward. This would not require additional experiments but a very careful analysis of the structures in the data bases and possible some sequence alignments to see how conserved the binding site is.

Reviewer #3:

Kasugamycin potentiates rifampicin in *Mycobacterium tuberculosis* by specifically decreasing mycobacterial mistranslation

Overall Comments:

This interesting, content-rich article, draws logical conclusions as to how kasugamycin reduces mistranslation and enhances the effects of rifampicin on *Mycobacterium tuberculosis* (Mtb). The detailed methods investigate these effects genotypically, phenotypically and using a murine model comparing the means in the effects of rifampicin +/- kasugamycin +/- streptomycin.

Presentation of data

This is clear, but due to the density of information and number of methods used, it might be useful to have clearly stated objectives and how these were achieved. For example, in the third objective: "Kasugamycin increases rifampicin susceptibility in vivo", it could be noted that this was achieved through pharmacokinetic characterization of Kasugamycin and streptomycin, assessing CFUs from lung and spleen tissue, and determining toxicity. Similarly, if the figures and tables are displayed according to the objectives, it would be easier to cross-reference.

Figures

Compressing the display information into 2 figures makes it difficult to read. Since up to 4 figures / tables are allowed, perhaps rearrange this information to enhance the text. Again, consider linking the displayed items to the objectives.

Methods

In the Materials and methods section, describing where different doses of Kasugamycin were used and how these were determined, would help clarify the figures and results.

Conclusions

The information related to Kasugamycin increasing the susceptibility of M tuberculosis to Rifampicin in vitro was convincing in both methods, results and interpretation. I had some concerns regarding the mouse model work and information relating to Figure 2 and related tables/figures.

If co-administration of Kasugamycin resulted in a 30-fold boosting of rifampicin killing of Mtb in mouse lungs (but was toxic), and Kasugamycin pre-treatment decreases rifampicin resistance, can the authors explain why sequential treatment with Kasugamycin and rifampicin in the murine model only occurred in spleen but not lungs? It is noted that this was not explained by differences in tissue PK, but are there any other possibilities?

I was also a bit surprised to see, in Figure 2C and the Supplementary file 2 that provides the statistical detail, that killing with both strep and ksg was significant but "strep vs ksg" was not different. In subsection Kasugamycin increases rifampicin susceptibility in vivo”, where the experiment is described, the aminoglycoside concentrations for the experiment appeared to have been chosen to show that the ksg-enhanced rif activity was not due to inhibition of protein synthesis alone (by aminoglycosides). If I understand the intention correctly, the concentration of streptomycin was chosen to demonstrate that such inhibition can be shown in the mouse model (at 7xMIC thus inhibitory). The concentration of Kasugamycin, conversely, was such that killing was not likely to occur (0.8xMIC, non-inhibitory). Yet killing is shown by both equally in Figure 2C. How can this be explained? Is the MIC for Kasugamycin correct? At 0.8xMIC no such direct killing should occur, without RIF?

I found it equally intriguing, looking at Figure 2C, that the Rif+Ksg effect was seen in spleen but not lung, which, as mentioned above, was not explained by drug levels. Yet, looking at the figure, it seems that Rif+Ksg were equally effective in both organs reducing counts by about 1.5 logs (eyeballing the figure). Rif alone and Rif+ Strep were much less effective in spleen than lung, helping the effect of the Rif+Ksg in spleen to statistical significance. There seem to be mixed messages in here that could be carved out a bit better.

Reviewer #4:

In the present study, Swarnava Chaudhuri and colleagues explore the therapeutic potential of kasugamycin in tuberculosis treatment. They use in vitro and in vivo cultures of the tuberculosis agent, *Mycobacterium tuberculosis*, to show that kasugamycin administration prevents *M. tuberculosis* growth and sensitizes *M. tuberculosis* to another drug, RNA polymerase inhibitor rifampicin. Finally, the authors use in vitro translation system and laboratory strains *M. tuberculosis* in an attempt to test their hypothesis that kasugamycin alters *M. tuberculosis* growth and rifampicin sensitivity by suppressing GatCAB-mediated mistranslation.

The current manuscript is unacceptable for publication due to the poor quality of its experimental design, non-justified conclusions and misleading content.

Firstly, kasugamycin has been extensively studied over the past few decades. It was shown that this drug is not simply a molecule that increases translation accuracy – as Chaudhuri and colleagues make the reader think, referring to (van Buul et al., 1984). Instead, kasugamycin was shown to:

1) Inhibit protein synthesis by suppressing translation initiation of canonical mRNAs (Okuyama et al., 1971, Poldermans et al., 1979, Moll et al., 2002, Schluenzen et al., 2006, Kaberdina et al., 2009, Surkov et al., 2010).

2) Allow translation of some leaderless mRNAs (Moll et al., 2002, Kaberdina et al., 2009, Lange et al., 2017).

3) Rapidly alter protein content of bacterial cells and trigger a complex stress-like response, which nature we still do not fully understand (Kaberdina et al., 2009, Muller et al., 2016, Lange et al., 2017).

Chaudhuri and colleagues totally disregard four decades of kasugamycin research, apart from (van Buul et al., 1984) and a structural study by (Schurwirth et al., 2006). Did the authors ignore the kasugamycin studies that disagree with kasugamycin as a specific inhibitor of mistranslation? Or is it due to ignorance about the major object of their study?

Secondly, the major conclusion of the manuscript – that kasugamycin inhibits *M. tuberculosis* by reducing Asn-to-Asp or Gln-to-Glu mistranslation – is not justified by their experimental data. I found no proof that kasugamycin specifically alters Asn-to-Asp or Gln-to-Glu mistranslation, because the read-through assay used in this study is not capable to discriminate Asn-to-Asp or Gln-to-Glu mistranslation from overall changes in the accuracy of protein synthesis. To prove that a small molecule alters one specific type of mistranslation (e.g. Asn-to-Asp and Gln-to-Glu but not other types of mistranslation) people typically use quantitative mass-spectrometry (for instance, Cvetesic et al., 2016).

Thirdly, I found several overstatements. For instance, the idea that "reducing mistranslation may be a novel mechanism for targeting bacterial adaptation" (the last sentence of the abstract) is by no means novel. It was pronounced in numerous papers by Paul Schimmel, Mike Ibba, Susan Martinis and others, including Babak Javid's group (e.g. Su et al., 2016).

[Editors’ note: what now follows is the decision letter after the authors submitted for further consideration.]

Thank you for resubmitting your work entitled "Kasugamycin potentiates rifampicin in *Mycobacterium tuberculosis* by specifically decreasing mycobacterial mistranslation" for further consideration at *eLife*. Your article has been favorably evaluated by two new reviewers together with Madhukar Pai (Reviewing Editor), and Gisela Storz (Senior Editor).

Based on the new evaluations, your article will be considered for publication in *eLife*. However, the issues raised by the new reviewers need to be addressed, as outlined below. Please address these comments as well as the comments of the original reviewers in a revised version of your manuscript. (We realize this is a little non-standard for *eLife* but your manuscript has taken an unusual path.)

Reviewer #5:

The revised manuscript by Chaudhuri et al., describes a synergistic role of kasugamycin with rifampicin to inhibit mycobacteria by reducing mistranslation. I feel this work could potentially be an important contribution to understanding the role of translational fidelity in bacteria-host interactions as well as developing a combinatory treatment for mycobacteria infections.

The major criticism of the previous submission comes from reviewer #4, and I share some of the same concerns. The authors convinced me that kasugamycin decreases the level of mistranslation in mycoplasma, which is supported by the in vivo and in vitro experiments using split and fused reporters. The challenge is to demonstrate that the potentiating effect of kasugamycin is due to reducing mistranslation. The results of the mutant strains with mutations in gatA and rpoB genes (Figure 1D, Figure 2C, and 2E) are particularly interesting and provide support for their conclusion. However, tolerance to antibiotics is very complicated and not fully understood. The use of an additional antibiotic or introducing a mutation in the RNA polymerase may change cellular responses (e.g., toxin/antitoxin levels or efflux) that lead to altered sensitivity to rifampicin. As the authors also rightfully admitted, other effects of kasugamycin cannot be ruled out. Kasugamycin inhibits translation initiation. I feel it would strengthen their conclusion by using other non-aminoglycoside antibiotics that inhibit initiation (e.g., thermorubin) and other steps of translation (Wilson, 2014) in the potentiating assays (Figure 2A). This would reveal if inhibiting translation in general has synergistic effects with rifampicin to limit the growth of mycobacteria. In this manuscript, the authors used streptomycin as a control, which enhances mistranslation itself and therefore is not a proper control for general translation inhibition.

To summarize, I feel this work is worth reconsidering to be published in *eLife* if the authors can provide further data (as suggested above and below) to strengthen the conclusion statement in the title. These experiments are not expected to take a long time.

Reviewer #6:

Despite being generally viewed as negative outcome, inaccurate translation of mRNA (commonly referred to as mistranslation) may in certain cases be beneficial for some organisms. One such case is *Mycobacterium tuberculosis*, in which mistranslation is related to emergence of rifampicin resistance. It has been proposed that mistranslation expands the phenotypic heterogeneity of the proteome, hence increasing the chances of antibiotic-resistant proteins. Based on this, Chaudhuri et al., hypothesize that decreasing mistranslation would lessen the occurrence of resistance. In this paper, the effect of kasugamycin on rifampicin resistance emergence is explored in pathogenic Mycobacterium strains. Using a fluorescence-based reporting system, the authors show that the addition of the antibiotic kasugamycin reduces indirect -tRNA mistranslation and prevents rifampicin-resistant strains emerging in vitro and in vivo. Finally, they are able to replicate these findings in a mouse model, in which they show that rifampicin killing potential is increased by addition of kasugamycin.

*Strengths in vitro…: The fluorescence-based assay used is this paper is an elegant and robust way of measuring the effects of mistranslation.

in vivo: The most interesting point of the paper, in my opinion, is how they try to translate their in vitro findings to a live model. Despite some mixed results (such as clearing the Mycobacterium in the spleen but not in the lungs) and the toxicity of the co-administration of kasugamycin and rifampicin, they show that these two drugs are able to partially clear the infection. This is a very exciting and promising finding and it would be interesting to see if they can improve and replicate this in humans in the future.

Assessment: The study of the "beneficial" effects of mistranslation is gaining a lot of attention. This study adds another piece of evidence about the link between translation accuracy and fitness. In this paper, the authors focus on the mistranslation product of the indirect pathway used for charging tRNA-Asn and tRNA-Gln. This is a very particular and well characterized "controlled" event of tRNA mischarging. The results presented in the first part of the paper (fluorescence-based assays and survival experiments) are adequate for the point the authors try to make, yet it is true that they are somewhat overused.

It is worth mentioning the effort the authors have placed to use a live mouse model to further test their findings. Physiology and pharmacokinetics are outside of my field of expertise, so I do not have anything meaningful to add on the second part of the study, but it seems promising. The authors seem to have problems delivering the antibiotics to high enough levels that do not cause toxicity as well, which casts some doubts about its possible application in humans.

*Weaknesses

Lack of alternative methods: the fluorescence-based method, despite its usefulness, is only an indirect way of detecting mistranslation events. Although I think that asking for additional, more direct experiments would be out of the scope of this work, it would be something to consider in the future.

---

## [Author Response]

[Editors’ note: the author responses to the first round of peer review follow.]

As you can see below, we received divergent feedback from the peer reviewers (reviewer #4, in particular, has raised several major concerns), and our final decision was reached after consultation between the reviewers. Based on these discussions and the individual reviews below, we regret to inform you that your work will not be considered further for publication in eLife.Reviewer #1:I am not a basic TB scientist, and my comments are focused more on the relevance of this study and its potential implications for TB treatment.Current TB treatment is long, and therapy for drug-resistant TB is particularly long, toxic, and expensive. Success rates are only about 50% and identification of new TB drug targets is a key priority for the TB field.In this novel study, the authors have identified kasugamycin as a small molecule that can specifically decrease mistranslation due to the indirect tRNA aminoacylation pathway. This, in turn could limit emergence of rifampicin resistance in vitro and increased mycobacterial susceptibility to rifampicin both in vitro and in a murine model of infection.The study opens a potential approach to saving rifampicin, which is one of the best TB drugs we have today. Of course, much more follow-up and clinical work in humans is necessary to follow up on this proof of concept work.As regards the lab methods, I will defer to expert reviewers working in this area.

We thank Dr. Pai for his positive comments regarding our work and its potential implications for salvaging rifampicin within the standard TB regimen.

Reviewer #2:Overall this is a very nice article trying to decipher the action of kasugamycin and its synergist effects with rifampicin. Considering the high death toll of Tuberculosis and the danger of MDR and XMDR Tb strains this is an important field of wide interest.

We thank the reviewer for their positive assessment of our work, that our conclusions appear to be justified and for stating that the data are of potentially wide interest.

In order to provide more context for the general reader I would suggest to expand the discussion on other approaches to combat MDR TBR from a mere list of papers to a concise explanation of alternative strategies, as little as one sentence may be sufficient. I couldn't help noticing that recent excellent work from the Baulard group (ie. Blondiaux et al., 2017) could be mentioned.

Thank you for this comment. We have now included a brief discussion of alternative strategies to rescue existing anti-TB drugs, including the recent important paper from Baulard and colleagues (see Discussion section).

The conclusions by the authors appear to be justified, however, I feel that the statistical treatment of a sometimes limited set of experiments needs to be explained in more detail. At several cases it is not clear how many independent measurements were taken to calculate the reported mean values and the error bars. I would suggest that the authors state that explicitly including the experiments in the supplementary material. I can only assume that the figures with individual data points refer to individual mice, so the authors should clarify that.

Thank you. We have now added additional details to the Materials and methods section and Figure legends to improve clarity of how the data were generated, including explicitly stated (as correctly surmised by the reviewer) that individual mouse organ bacterial burdens are presented in Figure 2.

Given that the paper is not very long, I would also suggest that the authors consider which of the figures on the supplementary material should be in the main text.

Thank you for this suggestion on how to improve the readability of our manuscript. The revised manuscript is now composed of three main figures, with some of the more important supplemental data moved to the main figures to assist the reader.

In order to provide more molecular insight I'd like to suggest that the authors discuss the actual binding site of kasugamycin in the ribosome. This may provide further evidence for the hypothesis put forward. This would not require additional experiments but a very careful analysis of the structures in the data bases and possible some sequence alignments to see how conserved the binding site is.

Thank you for this suggestion. There are two published structures of kasugamycin bound to the ribosome: Schuwirth et al., 2006, which was cited in the originally submitted manuscript, showing the structure of the E. coli ribosome and kasugamycin, and Schluenzen et al., 2006, which shows two potential kasugamycin binding sites in the *T. thermophilus* ribosome.

These sites are highly conserved in all bacteria, as is the resistance mechanism to

kasugamycin: kasugamycin action requires N6 methylation of two adjacent adenosine nucleotides (A1518 and A1519, E. coli numbering) in 16S rRNA, which is mediated by the KsgA methyltransferase. These prior data suggest that kasugamycin is likely to bind to the same primary site in all bacterial ribosomes. We have added a brief discussion of these points in the revised manuscript (see Discussion section).

Reviewer #3:Kasugamycin potentiates rifampicin in Mycobacterium tuberculosis by specifically decreasing mycobacterial mistranslation

We thank Dr. Diacon for his comments on our manuscript, in particular his assessment that our study is both logical in its deductions, and multi-layered in approach.

Specifically:

Overall Comments:This interesting, content-rich article, draws logical conclusions as to how kasugamycin reduces mistranslation and enhances the effects of rifampicin on Mycobacterium tuberculosis (Mtb). The detailed methods investigate these effects genotypically, phenotypically and using a murine model comparing the means in the effects of rifampicin +/- kasugamycin +/- streptomycin.Presentation of dataThis is clear, but due to the density of information and number of methods used, it might be useful to have clearly stated objectives and how these were achieved. For example, in the third objective: "Kasugamycin increases rifampicin susceptibility in vivo", it could be noted that this was achieved through pharmacokinetic characterization of Kasugamycin and streptomycin, assessing CFUs from lung and spleen tissue, and determining toxicity. Similarly, if the figures and tables are displayed according to the objectives, it would be easier to cross-reference.

Thank you for these comments to improve the readability of our manuscript. We have now revised the manuscript, explicitly stating our objectives at the beginning of each subsection of the Results section.

FiguresCompressing the display information into 2 figures makes it difficult to read. Since up to 4 figures / tables are allowed, perhaps rearrange this information to enhance the text. Again, consider linking the displayed items to the objectives.

Thank you. As also suggested by reviewer #2, the revised manuscript comprises three main figures.

MethodsIn the Materials and methods section, describing where different doses of Kasugamycin were used and how these were determined, would help clarify the figures and results.

Thank you for this suggestion. The Materials and methods section and/ or Figure legends now include this information including the rationale for dose selection.

ConclusionsThe information related to Kasugamycin increasing the susceptibility of M tuberculosis to Rifampicin in vitro was convincing in both methods, results and interpretation. I had some concerns regarding the mouse model work and information relating to Figure 2 and related tables/figures.If co-administration of Kasugamycin resulted in a 30-fold boosting of rifampicin killing of Mtb in mouse lungs (but was toxic), and Kasugamycin pre-treatment decreases rifampicin resistance, can the authors explain why sequential treatment with Kasugamycin and rifampicin in the murine model only occurred in spleen but not lungs? It is noted that this was not explained by differences in tissue PK, but are there any other possibilities?

a) Lung vs. Spleen Efficacy in alternate dosing model: We can only speculate regarding the reason for lack of efficacy in kasugamycin boosting of rifampicin in lungs at 1 month in the alternate dosing regimen. The results are consistent over several replications. Although measurement of bulk tissue PK (i.e. in homogenates) did not reveal any differences between lungs and spleens in kasugamycin dosing, such bulk measurements would not be able to detect any intra-tissue heterogeneity: as such, it is possible that kasugamycin and rifampicin in the alternate dosing schedule were targeting similar bacterial populations within spleens, but different bacterial populations within the lungs, potentially explaining the results. (see Discussion section).

I was also a bit surprised to see, in Figure 2C and the Supplementary file 2 that provides the statistical detail, that killing with both strep and ksg was significant but "strep vs ksg" was not different. In subsection Kasugamycin increases rifampicin susceptibility in vivo”, where the experiment is described, the aminoglycoside concentrations for the experiment appeared to have been chosen to show that the ksg-enhanced rif activity was not due to inhibition of protein synthesis alone (by aminoglycosides). If I understand the intention correctly, the concentration of streptomycin was chosen to demonstrate that such inhibition can be shown in the mouse model (at 7xMIC thus inhibitory). The concentration of Kasugamycin, conversely, was such that killing was not likely to occur (0.8xMIC, non-inhibitory). Yet killing is shown by both equally in Figure 2C. How can this be explained? Is the MIC for Kasugamycin correct? At 0.8xMIC no such direct killing should occur, without RIF?

b) Ksg vs Strep alone significance: For almost all anti-TB drugs, peak plasma concentrations at least 1-2 logs greater than in vitro MIC are required for in vivo efficacy (Mitchison, 2012, Pasipanodya et al., 2013). The streptomycin C_max_/MIC was ~7 and had no statistically significant efficacy. Although not measured by us, the standard rifampicin dose given typically achieves a C_max_/MIC of ~100. This makes the kasugamycin (alone) result even more remarkable. As correctly observed, the plasma C_max_ never reaches MIC, and the tissue-specific PK measurements suggest tissue concentrations of Ksg are approximately 30-40x *lower* than the in vitro MIC – although with more stable kinetics than plasma concentrations. We have now expanded our discussion of these findings, which although not the central focus of the current work, are nonetheless intriguing. Although purely speculation at this point, and the subject of future study, we believe that the kasugamycin alone is not acting as a typical antibiotic, but may be interfering with “adaptive mistranslation/ translation” of the *M. tuberculosis* (see also comments to reviewer # 4 below). This may explain why it is able to restrict *M. tuberculosis* growth without achieving MIC. Alternative explanations could be that kasugamycin gets massively concentrated in tissues (not the case from our bulk tissue PK studies), or in macrophages – although we haven’t measured macrophage concentrations of kasugamycin, for all aminoglycosides where it has been measured, aminoglycoside intra-cellular penetration is extremely poor, making this unlikely. Nonetheless, we have now included all these possibilities in our discussion.

We have now clarified our streptomycin dosing choice: we wanted to ensure that any effects of rifampicin+kasugamycin were not due to non-specific aminoglycoside translation inhibition or post-antibiotic effects. The maximum C_max_/MIC achievable with kasugamycin was 0.8. We measured the MIC of *M. tuberculosis*-H37Rv to kasugamycin several times and by different methods. These varied from 400-1000 µg/ml according to method, but we stated the most conservative (i.e. lowest MIC) measurement rather than a mean in the manuscript so as to avoid under-estimating the C_max_/ MIC of 0.8. The PK equivalent dosing of streptomycin would be 0.4mg/kg. However, we wished to not under-estimate streptomycin effects, therefore used a dose of 3mg/kg (C_max_/MIC of ~ 7): this would be within an order of magnitude (but higher) than the equivalent kasugamycin dose, but would not be so high as to confound rifampicin potentiation by simple additive effects of what is known to be a bactericidal anti-TB drug. Since streptomycin is fairly well tolerated, much higher doses (1560 mg/kg) have historically been used to achieve efficacy (equivalent to C_max_/MIC of 30120, i.e. 35-150x equivalent kasugamycin dose).

These points aside: the absolute lung and spleen bacterial burdens were 6.1 and 4.8 log_10_ CFU for streptomycin alone and 6.0 and 4.5 log_10_ CFU for kasugamycin alone (compared with no drug: 6.4 and 4.9 log_10_ CFU) respectively. The absolute reduction in lung and spleen bacterial burdens were 0.34 and 0.14 log_10_ CFU for streptomycin alone and 0.45 and 0.42 log_10_ CFU for kasugamycin alone (compared with no drug), suggesting greater efficacy of kasugamycin, but the difference between the two groups was not statistically significant. Nonetheless, kasugamycin alone (at 9x lower effective dose than streptomycin) but not streptomycin alone did achieve significance threshold compared with the no drug group.

I found it equally intriguing, looking at Figure 2C, that the Rif+Ksg effect was seen in spleen but not lung, which, as mentioned above, was not explained by drug levels. Yet, looking at the figure, it seems that Rif+Ksg were equally effective in both organs reducing counts by about 1.5 logs (eyeballing the figure). Rif alone and Rif+ Strep were much less effective in spleen than lung, helping the effect of the Rif+Ksg in spleen to statistical significance. There seem to be mixed messages in here that could be carved out a bit better.

The reviewer is correct that the relative magnitude of the RIF+Ksg effect was approximately 1.5 log CFU reduction in both lungs and spleens. However, the lack of significance in the lungs is due to the fact the RIF alone had almost identical efficacy in the lungs, but was significantly less effective in the spleens. It’s possible that rifampicin’s superior efficacy in the lungs is that it’s intra-organ distribution within lungs better matches where *M. tuberculosis* reside, or that there is a greater mismatch between rifampicin/ kasugamycin distribution within lungs compared with spleens, and these are now discussed in the revised manuscript.

Reviewer #4:In the present study, Swarnava Chaudhuri and colleagues explore the therapeutic potential of kasugamycin in tuberculosis treatment. They use in vitro and in vivo cultures of the tuberculosis agent, Mycobacterium tuberculosis, to show that kasugamycin administration prevents M. tuberculosis growth and sensitizes M. tuberculosis to another drug, RNA polymerase inhibitor rifampicin. Finally, the authors use in vitro translation system and laboratory strains M. tuberculosis in an attempt to test their hypothesis that kasugamycin alters M. tuberculosis growth and rifampicin sensitivity by suppressing GatCAB-mediated mistranslation.The current manuscript is unacceptable for publication due to the poor quality of its experimental design, non-justified conclusions and misleading content.

We thank the reviewer for taking the time to read our manuscript and their observations, which we have addressed below with the aim of improving the readability and interpretation of our data.

Firstly, kasugamycin has been extensively studied over the past few decades. It was shown that this drug is not simply a molecule that increases translation accuracy – as Chaudhuri and colleagues make the reader think, referring to (van Buul et al., 1984). Instead, kasugamycin was shown to:1) Inhibit protein synthesis by suppressing translation initiation of canonical mRNAs (Okuyama et al., 1971, Poldermans et al., 1979, Moll et al., 2002, Schluenzen et al., 2006, Kaberdina et al., 2009, Surkov et al., 2010).2) Allow translation of some leaderless mRNAs (Moll et al., 2002, Kaberdina et al., 2009, Lange et al., 2017).3) Rapidly alter protein content of bacterial cells and trigger a complex stress-like response, which nature we still do not fully understand (Kaberdina et al., 2009, Muller et al., 2016, Lange et al., 2017).Chaudhuri and colleagues totally disregard four decades of kasugamycin research, apart from (van Buul et al., 1984) and a structural study by (Schurwirth et al., 2006). Did the authors ignore the kasugamycin studies that disagree with kasugamycin as a specific inhibitor of mistranslation? Or is it due to ignorance about the major object of their study?

We thank the reviewer for these important observations. Aminoglycosides are familiar as a chemical group to both TB biologists (the first anti-TB medication was streptomycin) and to translation biologists (as translation inhibitors), the two groups most likely to have interest in our findings. We made reference to both these general properties of aminoglycosides, including kasugamycin and streptomycin in our manuscript. We chose to emphasise kasugamycin’s previously described role in reducing ribosomal misreading errors, since it was relevant to our rationale and this property may have been less familiar to most readers. As rightly pointed out, we apologise that we did not more fully cite some of the seminal prior work on kasugamycin’s other described activities in our manuscript. We have now expanded our paper and discussion of kasugamycin considerably, including discussion of some of these important papers (see Results section and Discussion section).

However, although these prior data (all performed in *E. coli* except for one crystal structure) do point to additional functions of kasugamycin, we believe that the rifampicin potentiation effects in particular, which is the central focus of our current study, are specifically and at least for the most part mediated by kasugamycin’s actions in reducing mistranslation due to the indirect pathway.

This is because:

a) In Figure 1—figure supplement 5 of the original manuscript (now Figure 2E), we utilised a strain we had previously described (in Su et al., 2016), *M. smegmatis*-RpoB-N434T (N434T). This strain has a single point mutation in the *rpoB* gene, altering an asparagine residue critical for rifampicin binding, to threonine. Aspartate, but not asparagine or threonine at residue 434 disrupts rifampicin binding to RNAP, and in our prior published work we showed by use of this strain and other genetic tools that mistranslation of this ASN residue in wild-type mycobacteria to ASP resulted in significant rifampicin tolerance. The N434T variant of RpoB can still bind rifampicin, but the threonine residue can no longer be mistranslated by the indirect tRNA aminoacylation pathway to aspartate, by definition, since mistranslation via that pathway is specific and limited to aspartate for asparagine and glutamate for glutamine misincorporations. This strain is more susceptible to rifampicin killing but is relatively resistant to kasugamycin potentiation of rifampicin.

The reported alternative actions of kasugamycin (e.g. on specific inhibition of translation of canonical but not leaderless mRNAs, non-specific stress responses etc) would not be altered by this single amino acid substitution. These data strongly argue that the rifampicin potentiation effects of kasugamycin, for the most part, are mediated by inhibition of mistranslation of this residue, which physiologically occurs in mycobacteria, due to the indirect tRNA aminoacylation pathway.

b) Furthermore, all of the effects in *M. smegmatis* and *M. tuberculosis* on potentiation of rifampicin in vitro occur at concentrations of kasugamycin between 30-150 µg/ml, when kasugamycin has no anti-bacterial activity in vitro. In data that I share below, but which was not included in the originally submitted manuscript, we tested two further activities of kasugamycin in *M. smegmatis* and failed to find evidence of either, even up to concentrations of 1500-2000 µg/ml kasugamycin.

Moll and colleagues and others (Moll et al., 2002, Kaberdina et al., 2009 and others) described that kasugamycin specifically inhibited translation of canonical mRNAs, but not leaderless transcripts. Of note, a large proportion of mycobacterial transcripts are leaderless. To investigate whether kasugamycin could inhibit canonical but not leaderless translation in mycobacteria, we constructed an *M. smegmatis* strain expressing GFP and mCherry from the same basic promoter (Psmyc), but GFP from a canonical version of the promoter and mCherry from a leaderless version of the promoter. 5’RACE verified the transcription start sites of the two promoters as described. At 1500 µg/ml kasugamycin, fully 10-50 times the concentration required to potentiate rifampicin or to decrease mistranslation, kasugamycin only slightly slowed bacterial growth, and failed to inhibit translation of either fluorescent protein (See Author response image 1 and Figure 1—figure supplement 2 in the revised manuscript).

**Author response image 1. respfig1:** Kasugamycin does not significantly inhibit translation of a canonical or leaderless transcript in *M. smegmatis* at doses much higher than required to reduce mistranslation. A strain of *M. smegmatis* was transformed with an episomal plasmid expressing *gfp* from the promoter *Psmyc* (canonical promoter with 5’ UTR) and a leaderless version of *Psmyc* driving *mCherry*. Chloramphenicol inhibited translation of both transcripts, whereas kasugamycin at 1500 µg/ml failed to significantly attenuate translation from either transcript.

In Kaberdina et al., 2009 Moll and colleagues showed that kasugamycin treatment in vitro resulted in intriguing 61S alternate ribosomes. We investigated whether we could identify such specialised ribosomes in *M. smegmatis*, and despite numerous attempts and different kasugamycin concentrations and time points, failed to do so (see Author response image 2 as a representative example), suggesting that mycobacteria may not form these specialised ribosomal structures.

**Author response image 2. respfig2:** Kasugamycin does not cause formation of 61S mycobacterial ribosomes. Ribosomal profiles of *M. smegmatis* ribosomes generated by sucrose gradient density centrifugation +/- kasugamycin treatment prior to isolation.

Whilst we are confident that the rifampicin potentiation effects of kasugamycin are due to its effects on mistranslation for the reasons cited in (a) above, we cannot 100% exclude other mechanisms mediating the kasugamycin only effect, which was only witnessed in vivo and is not the main focus of this paper. Although we have no evidence for these alternative mechanisms at play in mycobacteria, absence of evidence does not fully imply evidence of absence, therefore we have extended our discussion of the kasugamycin only phenotype to include the possibility that these other mechanisms may potentially play a role.

Secondly, the major conclusion of the manuscript – that kasugamycin inhibits M. tuberculosis by reducing Asn-to-Asp or Gln-to-Glu mistranslation – is not justified by their experimental data. I found no proof that kasugamycin specifically alters Asn-to-Asp or Gln-to-Glu mistranslation, because the read-through assay used in this study is not capable to discriminate Asn-to-Asp or Gln-to-Glu mistranslation from overall changes in the accuracy of protein synthesis. To prove that a small molecule alters one specific type of mistranslation (e.g. Asn-to-Asp and Gln-to-Glu but not other types of mistranslation) people typically use quantitative mass-spectrometry (for instance, Cvetesic et al., 2016).

To address this issue, which is, as the reviewer rightly surmises, central to the arguments of the manuscript, we would ask that the reviewer re-examines the data that we resummarise below, which were all included in the original submission.

a) First, we made no claims that kasugamycin only decreases mistranslation due to the indirect tRNA aminoacylation pathway and apologise if this was not communicated clearly enough. In fact, we cited the 1984 van Buul et al., paper to specifically address that kasugamycin had been previously implicated in decreasing errors in ribosomal misreading. However, no previous studies (to our knowledge) have ever examined the effects of kasugamycin in decreasing errors in the indirect tRNA aminoacylation pathway, which as the Reviewer mentions, is via potentially an entirely different mechanism.

b) To specifically address whether kasugamycin decreases mistranslation via the indirect tRNA aminoacylation pathway (a central claim of the paper), we used reporters that measure misincorporation (not readthrough) of asparagine to aspartate (or glutamate for glutamine in one of the Ren-FF reporters). A dual Renilla-Firefly luciferase reporter system has been characterised in detail in prior publications (Javid et al., 2014 and Su et al., 2016). The Nluc/GFP reporter is used for the first time here. Both sets of reporters work via the same principle. Both Nluc and Renilla luciferases have aspartate residues (D140 in Nluc, and D120 in Renilla) critical for enzyme activity. Mutation of these residues to asparagine results in 2-3 orders of magnitude loss of function. Substitution of the coded asparagine (AAC codon) for aspartate during translation by translational error would result in gain in function of the mistranslated subpopulation of luciferase enzyme, which can be sensitively detected. The reporter by itself cannot formally distinguish whether substitution of asparagine for aspartate is via misincorporation of misacylated tRNA (Asp-tRNA^Asn^) or ribosomal misreading of AAC codon for GAC (Asp) – although we have previously demonstrated (in Leng et al., 2015), that codon position 1 ribosomal misreading errors in mycobacteria occur at far lower frequencies than the aspartate for asparagine substitution error rates shown here and in Su et al. 2016. It should be noted that any other detection method of amino acid substitution (e.g. the suggested mass spectrometry route) would not be able to distinguish the source of mistranslation either. However, we have two strong lines of evidence that together unequivocally suggest that kasugamycin is able to decrease mistranslation of the indirect pathway i.e. misincorporation of misacylated tRNA (AsptRNA^Asn^) at AAC codons:

i) In Su et al., 2016 we characterised strains of *M. smegmatis* with partial loss of function in GatCAB (the key enzyme in the indirect pathway) due to mutations in *gatA*. In that work we showed using *gatA* mutations on an otherwise isogenic background, as well as genetic complementation, that mutations in *gatA* were sufficient for significantly increasing mistranslation of aspartate for asparagine/ glutamate for glutamine, and that in the mutant strains, the defect in translational fidelity is specifically in the indirect tRNA aminoacylation pathway (see Figure 1 and Figure 1—figure supplement 2 in that work).

In this work, we use kasugamycin to measure the change in mistranslation rate in strain HWS.4 (*gatA*-V405D), which, as described above, has an increased mistranslation rate solely due to a defect in the indirect tRNA aminoacylation pathway. We show that kasugamycin can significantly reduce the mistranslation rate in this strain by a greater degree than the total mistranslation rate of the corresponding parent strain (in absolute terms, by > 4%/ codon, which is higher than the total mistranslation rate in the wild-type mycobacterial strain under the same conditions). It should be noted that in other data from Su et al., 2016, we showed that even in wild-type mycobacteria, the major source of this specific type of error is due to the indirect tRNA aminoacylation pathway, which provided the rationale for using kasugamycin to target mistranslation in wild-type *M. tuberculosis*. See Figure 1D of the revised manuscript. Our interpretation of these data, are that the reduction in mistranslation as measured by the reporter, must at least partly (and we would argue for the most part) be due to reduction in mistranslation due to the indirect tRNA aminoacylation pathway.

ii) The hybrid cell-free translation system provides the strongest evidence that kasugamycin can decrease error that is generated from misacylated Asp-tRNA^Asn^ (Figure 1D in the original manuscript now 1E). This *E. coli* translation systems lacks the indirect pathway. The corrected Nluc activity (as measured by Nluc/ GFP – and a sensitive measure of misincorporation, by any mechanism, of aspartate for asparagine) is low but measurable (since the abrogation of Nluc activity in the D140N substitution results in 100-fold loss of activity, the high potency of the Nluc enzyme means that residual activity can still be measured sensitively). A non-discriminatory aspartyl synthetase specifically misacylates tRNA^Asn^ to Asp-tRNA^Asn^, with no opportunity for correction due to lack of GatCAB in the system, and addition of this enzyme to the cell-free translation system increases the reporter measured Nluc activity (i.e. mistranslation error rate) three-fold. The only source of increased error in this system is misacylated Asp-tRNA^Asn^. Kasugamycin, at doses that *do not inhibit translation of GFP in the Nluc-GFP fusion protein*, specifically decrease the measured gain in Nluc activity in a dose-dependent manner to baseline. These data strongly and unequivocally argue that kasugamycin is capable of increasing fidelity against mistranslation generated by the indirect tRNA aminoacylation pathway.

Although mass-spectrometry is sensitive at detecting many misincorporations, the detection of aspartate for asparagine or glutamate for glutamine pose specific technical challenges. First, the mass shift of these two mistranslation events is 1 Dalton, which makes it difficult to distinguish mistranslation events of a small (typically 1%) subpopulation of otherwise identical peptides from the naturally occurring isotope envelopes in the spectra. Secondly, and more importantly, sample preparation for mass spectrometry routinely encounters high (5%+/ asparagine or glutamine residue) post-lysis deamidation (see e.g. PMID 1678690) which is identical to the mistranslation that occurs in the indirect pathway. As such, measurement of physiological mistranslation due to the indirect pathway is technically challenging by mass spectrometry and at any rate, as mentioned above, detection of aspartate for asparagine substitutions by mass spectrometry would not be able to infer the molecular mechanistic source of the substitution. We agree that mass spectrometry would be able to detect other potentially easier to detect substitutions, but we have not argued that kasugamycin solely increases discrimination of errors from the indirect pathway. We have argued that kasugamycin’s activity in reducing error in this pathway is responsible for the potentiation of rifampicin killing for the reasons outlined in point (1a) above.

We stand by our claim that our data supports the interpretation that kasugamycin can increase discrimination of misacylated tRNAs generated by the indirect pathway. We do not claim that kasugamycin does so exclusively and does not increase fidelity of translation due to ribosomal misreading, and we have added a note in the Discussion to address this point specifically.

Thirdly, I found several overstatements. For instance, the idea that "reducing mistranslation may be a novel mechanism for targeting bacterial adaptation" (the last sentence of the abstract) is by no means novel. It was pronounced in numerous papers by Paul Schimmel, Mike Ibba, Susan Martinis and others, including Babak Javid's group (e.g. Su et al., 2016)

Thank you for your comment. Of course, and as clearly stated throughout the manuscript, the rationale for identifying a small molecule that targets bacterial mistranslation was provided by both our own prior work (specifically in the mycobacterial system) as well as the work of Ibba, Schimmel, Martinis and others in other bacterial systems (all cited and discussed in our prior review, which we cite in the manuscript: Ribas de Pouplana et al., 2014). However, to our knowledge, all prior work had been genetic validation of the principles of “adaptive mistranslation”. We therefore believe the discovery that bacterial adaptive mistranslation is amenable to pharmacological targeting is both novel, and an important proof of principle, and therefore worth emphasising, in the hope that our work spurs further research in this area. We have, however, changed the wording to “pharmacologically reducing mistranslation may be a novel mechanism for targeting bacterial adaptation” in order to be explicit in our intention.

[Editors' note: the author responses to the re-review follow.]

Based on the new evaluations, your article will be considered for publication in eLife. However, the issues raised by the new reviewers need to be addressed, as outlined below. Please address these comments as well as the comments of the original reviewers in a revised version of your manuscript. (We realize this is a little non-standard for eLife but your manuscript has taken an unusual path.)

I would like to thank you as Senior Editor, and Dr. Madhukar Pai as Reviewing Editor of our recently submitted manuscript to *eLife* entitled “Kasugamycin potentiates rifampicin and limits emergence of resistance in *Mycobacterium tuberculosis* by specifically decreasing mycobacterial mistranslation”. We appreciate the opportunity to revise our manuscript in light of comments from additional reviewers following the initial resubmission.

Reviewer #5:The revised manuscript by Chaudhuri et al. describes a synergistic role of kasugamycin with rifampicin to inhibit mycobacteria by reducing mistranslation. I feel this work could potentially be an important contribution to understanding the role of translational fidelity in bacteria-host interactions as well as developing a combinatory treatment for mycobacteria infections.

We thank the reviewer for their assessment of our work as making a potentially important contribution in understanding the role of translational fidelity in host-pathogen interactions. With regards to specific comments raised:

The major criticism of the previous submission comes from reviewer #4, and I share some of the same concerns. The authors convinced me that kasugamycin decreases the level of mistranslation in mycoplasma, which is supported by the in vivo and in vitro experiments using split and fused reporters. The challenge is to demonstrate that the potentiating effect of kasugamycin is due to reducing mistranslation. The results of the mutant strains with mutations in gatA and rpoB genes (Figures 1D, 2C, and 2E) are particularly interesting and provide support for their conclusion. However, tolerance to antibiotics is very complicated and not fully understood. The use of an additional antibiotic or introducing a mutation in the RNA polymerase may change cellular responses (e.g., toxin/antitoxin levels or efflux) that lead to altered sensitivity to rifampicin. As the authors also rightfully admitted, other effects of kasugamycin cannot be ruled out. Kasugamycin inhibits translation initiation. I feel it would strengthen their conclusion by using other non-aminoglycoside antibiotics that inhibit initiation (e.g., thermorubin) and other steps of translation (Wilson, 2014) in the potentiating assays (Figure 2A). This would reveal if inhibiting translation in general has synergistic effects with rifampicin to limit the growth of mycobacteria. In this manuscript, the authors used streptomycin as a control, which enhances mistranslation itself and therefore is not a proper control for general translation inhibition.

Thank you for these comments. There are four well-characterised 30S translation initiation inhibitors in the literature (as in e.g. the Wilson review, cited above), being: edeine, thermorubin, pactamycin and kasugamycin. Of these, only kasugamycin is currently commercially available. We were unable to identify commercial sources of any of the other three inhibitors, in China, the US or the UK, and none of the 6 or so chemistry CROs that we approached in China were willing or able to synthesise any of the three reagents for us. Nonetheless, we eventually identified Prof. Ian Brierley at the University of Cambridge, who has published on edeine in cell-free translation systems. He had purchased a batch of edeine >> 5 years ago when it was still commercially available, and he generously provided us with a small quantity of the reagent for evaluation. We have now tested Edeine, as well as chloramphenicol, an inhibitor of peptide bond formation in both the mistranslation assay (Figure 1—figure supplement 3) and rifampicin potentiating assay (Figure 2—figure supplement 2). Intriguingly, at subMIC concentrations, Edeine also decreases mistranslation rates and potentiates rifampicin, but chloramphenicol does not. These data suggest that inhibitors of 30S initiation, but not other translation inhibitors may increase ribosomal discrimination of misacylated Asp-tRNAAsn and Glu-tRNAGln during translation. We have now added a note in the Discussion section regarding these observations.